# FARI: ROBUST ONE-STEP INVERSION FOR WATERMARKING IN DIFFUSION MODELS

**Jindong Yang**[1,2], **Han Fang**[1,2*], **Weiming Zhang**[1,2], **Nenghai Yu**[1,2], **Kejiang Chen**[1,2*]
[1]University of Science and Technology of China
[2]Anhui Province Key Laboratory of Digital Security
`dx929@mail.ustc.edu.cn`
`{ynh, zhangwm, chenkj, fanghan}@ustc.edu.cn`

## ABSTRACT

Inversion-based watermarking is a promising approach to authenticate diffusion-generated images, yet practical use is bottlenecked by inversion that is both slow and error-prone. While the primary challenge in the watermarking setting is robustness against external distortions, existing approaches over-optimize internal truncation error, and because that error scales with the sampler step size, they are inherently confined to high-NFE (number of function evaluations) regimes that cannot meet the dual demands of speed and robustness. In this work, we have two key observations: (i) the inversion trajectory has markedly lower curvature than the forward generation path does, making it highly compressible and amenable to low-NFE approximation; and (ii) in inversion for watermark verification, the trade-off between speed and truncation error is less critical, since external distortions dominate the error. A faster inverter provides a dual benefit: it is not only more efficient, but it also enables end-to-end adversarial training to directly target robustness, a task that is computationally prohibitive for the original, lengthy inversion trajectories. Building on this, we propose **FARI** (**F**ast **A**symmetric **R**obust **I**nversion), a one-step inversion framework paired with lightweight adversarial LoRA fine-tuning of the denoiser for watermark extraction. While consolidation slightly increases internal error, FARI delivers large gains in both speed and robustness: with 20 minutes of fine-tuning on a single NVIDIA RTX A6000 GPU, it surpasses 50-step DDIM inversion on watermark-verification robustness while dramatically reducing inference time.

## 1 INTRODUCTION

The rapid proliferation of diffusion models (Ho et al., 2020; Song et al., 2021) has led to an explosion of AI-generated content, simplifying creative production but also fueling the spread of synthetic misinformation and raising concerns about intellectual property protection for model providers. In response, inversion-based watermarking (Yang et al., 2024; Wen et al., 2023; Huang et al., 2024; Gunn et al., 2024) has shown remarkable promise for authenticating and tracing diffusion-generated images. By embedding a watermark in the initial noise, the mark becomes deeply integrated with the image's semantics during the iterative generation process, ensuring minimal visual impact. To extract the watermark, the image typically needs to be reconstructed back to noise via inversion techniques (Song et al., 2021; Hong et al., 2024). This inversion step, however, is the method's critical bottleneck. It is computationally expensive, time-consuming, and introduces substantial errors, all of which hinder the practical, large-scale deployment of inversion-based watermarks. This bottleneck motivates the development of a fast and accurate inversion method tailored for watermark extraction.

Although many inversion techniques (Mokady et al., 2023; Hong et al., 2024; Wallace et al., 2023) exist, most primarily aim to reduce internal inversion error (caused by discretization truncation and Classifier-Free Guidance) via iteration (Pan et al., 2023; Garibi et al., 2024; Samuel et al., 2023), optimization (Hong et al., 2024; Li et al., 2024; Mokady et al., 2023), or analytical control of

---

*Corresponding authors.

truncation-error bounds (Wallace et al., 2023; Zhang et al., 2024a; Wang et al., 2024). While effective for diffusion-based image editing (Hertz et al., 2022), this strategy is ill-suited for watermarking. Prior to extraction, images may be subjected to diverse distortions (e.g., JPEG compression, blur) that induce substantial initial-condition shifts; given the denoiser's sensitivity, these perturbations compound rapidly along the inversion trajectory and become the dominant bottleneck to extraction accuracy.

This shift in the error bottleneck for watermark extraction leads us to question the necessity of traditional high-NFE inversion. In prior methods, which primarily address clean-image scenarios, performance is limited by the discretization truncation error of the ODE sampler, which is directly related to the step size. Consequently, a high NFE is required to maintain precision. In the watermarking context, however, this internal error is dwarfed by the accumulated error from external distortions, a factor that is not explicitly mitigated by a larger number of steps. Furthermore, the natural solution to instill this robustness, adversarial training, is blocked by the high NFE of these traditional inverters. A powerful end-to-end training regime is rendered computationally infeasible by the prohibitive memory costs of backpropagating through a long iterative process. Meanwhile, the more computationally feasible, factorized objective, similar to that used in diffusion pretraining (Ho et al., 2020), proves insufficient for learning the global robustness required to counter complex distortions (see Appendix E.3). These facts indicate that first finding a low-NFE solution is not only beneficial for speed, but also enables a breakthrough in enhancing robustness.

Motivated by this, we propose **FARI**: **F**ast **A**symmetric **R**obust **I**nversion, a framework that achieves fast and robust inversion tailor-made for watermarking at a minimal fine-tuning cost. FARI is based on a key insight into the geometric asymmetry between generation and inversion trajectories: while the estimation error in inversion makes the reconstructed noise inaccurate, it also indirectly endows the inversion path with a significantly lower curvature than its generation counterpart. A lower-curvature trajectory is inherently easier to approximate with fewer steps. This enables a step-distillation approach that collapses multi-step inversion into a single efficient step. This reduction in NFE unlocks efficient end-to-end adversarial training. While this distillation-based estimation slightly sacrifices precision on clean, distortion-free inversion, the direct speed-up and the indirect enhancement in robustness are substantial, and we find that the downside of this trade-off has a negligible effect on the performance of the downstream watermarking task (Yang et al., 2024; Wen et al., 2023). Furthermore, our use of LoRA (Hu et al., 2022) for fine-tuning elegantly avoids the degradation of image quality. By storing the learned robustness knowledge externally in the LoRA parameters, we can simply deactivate the LoRA branch during generation, ensuring that the original model's generation quality remains unchanged. Our experiments demonstrate that with just 20 minutes of fine-tuning on a single NVIDIA RTX A6000 GPU, the one-step FARI surpasses the robustness of the 50-step DDIM baseline in watermark verification tasks.

## 2 BACKGROUND

### 2.1 DIFFUSION MODELS

Diffusion models (Ho et al., 2020; Song et al., 2021) are a class of generative models that operate by iteratively transforming a pure Gaussian noise vector $z_T \sim \mathcal{N}(0, I)$ into a real data sample $z_0 \sim q(z)$ through $T$ denoising steps. The process is defined by two Markov chains. The forward process gradually diffuses a data sample $z_0$ by adding Gaussian noise over $T$ timesteps according to a fixed variance schedule $\{\beta_t\}_{t=1}^T$:

$$q(z_t|z_{t-1}) = \mathcal{N}(z_t; \sqrt{1-\beta_t}z_{t-1}, \beta_t I), \tag{1}$$

A key property of this process is that we can sample $z_t$ at any arbitrary timestep $t$ directly from $z_0$:

$$z_t = \sqrt{\bar{\alpha}_t}z_0 + \sqrt{1-\bar{\alpha}_t}\epsilon, \tag{2}$$

where $\alpha_t = 1 - \beta_t$, $\bar{\alpha}_t = \prod_{i=1}^t \alpha_i$, and $\epsilon \sim \mathcal{N}(0, I)$. The reverse process learns to denoise these corrupted samples to recover the original data. This is achieved by training a neural network $\epsilon_\theta$ to predict the added noise $\epsilon$ from the noisy input $z_t$. The objective function is typically a simplified version of the evidence lower bound:

$$\mathcal{L}(\theta) = \mathbb{E}_{z_0, t\sim\text{Uniform}(1,T), \epsilon\sim\mathcal{N}(0,I)} \left[\|\epsilon - \epsilon_\theta(z_t, t)\|_2^2\right], \tag{3}$$

## 2.2 DDIM SAMPLING AND INVERSION

The denoising diffusion implicit model (Song et al., 2021) (DDIM) provides a deterministic sampling process by defining a non-Markovian forward process that leads to the same marginal distributions. Given a noisy latent $z_t$, DDIM computes the subsequent latent $z_{t-1}$ by first predicting an estimate of the clean image, $\hat{z}_0$, and then stepping towards it:

$$\hat{z}_0 = \frac{z_t - \sqrt{1 - \bar{\alpha}_t}\epsilon_\theta(z_t, t)}{\sqrt{\bar{\alpha}_t}}, \tag{4}$$

$$z_{t-1} = \sqrt{\bar{\alpha}_{t-1}}\hat{z}_0 + \sqrt{1 - \bar{\alpha}_{t-1}}\epsilon_\theta(z_t, t). \tag{5}$$

The deterministic nature of DDIM is crucial as it allows for an invertible generation process, which iteratively computes $z_t$ from $z_{t-1}$ by reversing the sampling steps. This unique invertible characteristic allows us to recover the initial noise representation $z_T$ from any generated image $z_0$, which serves as a powerful tool for inversion-based watermarking.

## 2.3 INVERSION-BASED WATERMARKING FOR DIFFUSION MODELS

We categorize these methods into three classes. The first class, epitomized by Tree-Ring (Wen et al., 2023), embeds a robust pattern into the Fourier domain of the initial noise to enable detection. Subsequent works have focused on enhancing its practical applications or extending its capabilities. For instance, RingID (Ci et al., 2024) extends it to a multi-bit watermark, ROBIN (Huang et al., 2024) improves its imperceptibility, and ZoDiac (Zhang et al., 2024b) generalizes it as a post-processing watermark, all without altering the core embedding and extraction logic. The second class, represented by Gaussian Shading (Yang et al., 2024), embeds a multi-bit watermark into the spatial domain of the noise through distribution-preserving sampling. Follow-up research has concentrated on improving its key reuse problem, as seen in PRC-Watermark (Gunn et al., 2024) and Gaussian Shading++ (Yang et al., 2025), and on functional extensions; for example, TAG-WM (Chen et al., 2025) and VideoShield (Hu et al., 2025) provide functionality for detecting tampered regions. The third class, such as GaussMarker (Li et al., 2025), combines the first two approaches to compensate for their weakness against geometric distortions.

## 2.4 INVERSION METHODS

There is a substantial body of work on diffusion model inversion. Methods such as BELM (Wang et al., 2024), BDIA (Zhang et al., 2024a), and EDICT (Wallace et al., 2023) directly modify the sampling process to make it invertible. Others, including AIDI (Pan et al., 2023), ExactDPM (Hong et al., 2024), and ReNoise (Garibi et al., 2024), employ iteration or gradient descent to obtain better intermediate values for trajectory alignment. A third category, which includes NTI (Mokady et al., 2023) and NPI (Miyake et al., 2025), focuses on optimizing a better null-text embedding to guide the regeneration process. As we have previously mentioned, these methods are primarily designed for training-free image editing. Consequently, they may fail in adversarial watermark extraction scenarios, a point we will demonstrate in our experiments section.

## 2.5 DIFFUSION MODEL ACCELERATION

The acceleration of diffusion models can be broadly categorized into two paths. The first path involves using solvers with lower truncation error (Lu et al., 2022a;b; Zhang & Chen, 2022). While these methods can reduce the number of inference steps to between 20 and 30, the quality of image generation in extreme few-step scenarios (e.g., $< 10$) remains unsatisfactory. A noteworthy method in this category is the AMED-Solver (Zhou et al., 2024a), which is based on the mean value theorem. It uses a small model to predict the timestep where the mean value occurs, thereby estimating the average velocity and enabling generation in as few as two steps. The second path is distillation, where a student model is trained to replicate the output of multiple teacher steps in a single step. Techniques such as progressive distillation (Salimans & Ho, 2022), consistency distillation (Song et al., 2023), and distribution matching distillation (Yin et al., 2024b;a) follow this paradigm. However, these methods typically demand a substantial amount of pre-generated training data, GPU memory, and time.

# 3 OUR PROPOSED METHOD

In this section, we propose FARI, a robust one-step inversion method designed for watermark extraction. It is based on our key finding that the inversion trajectory exhibits a significantly lower curvature than the generation path does, enabling efficient one-step distillation, which in turn makes the adversarial fine-tuning computationally feasible.

## 3.1 THE INVERSION TRAJECTORY EXHIBITS LOWER CURVATURE

We begin with the inherent systematic error in DDIM inversion (Song et al., 2021). For deterministic DDIM sampling, the denoising process, which computes $z_{t-1}$ from $z_t$, can be written in a single recurrence relation(the conditioning terms are omitted for simplicity):

$$z_{t-1} = \sqrt{\frac{\bar{\alpha}_{t-1}}{\bar{\alpha}_t}} z_t + \left( \sqrt{1 - \bar{\alpha}_{t-1}} - \sqrt{\frac{\bar{\alpha}_{t-1}(1 - \bar{\alpha}_t)}{\bar{\alpha}_t}} \right) \epsilon_\theta(z_t, t). \tag{6}$$

The inversion step, which solves for $z_t$ based on $z_{t-1}$, is derived as:

$$z_t = \sqrt{\frac{\bar{\alpha}_t}{\bar{\alpha}_{t-1}}} z_{t-1} + \left( \sqrt{1 - \bar{\alpha}_t} - \sqrt{\frac{\bar{\alpha}_t(1 - \bar{\alpha}_{t-1})}{\bar{\alpha}_{t-1}}} \right) \epsilon_\theta(z_t, t). \tag{7}$$

However, since our goal is to solve for $z_t$, the term $\epsilon_\theta(z_t, t)$ on the right-hand side of Eq. 7 cannot be explicitly calculated. Generally, this is addressed by making a piecewise linear assumption, approximating $\epsilon_\theta(z_t, t) \approx \epsilon_\theta(z_{t-1}, t)$. The validity of this assumption, however, requires a sufficiently small step size, a condition that practical settings often fail to meet. This becomes a significant source of inversion error, even for clean images. While many works (Lin et al., 2024; Wang et al., 2024; Staniszewski et al., 2024) have recognized that changes in the trajectory direction $\epsilon_\theta(\cdot)$ cause an offset of the reconstructed noise $\hat{z}_T$ and have attempted to mitigate this asymmetry, we further point out that under the combined effect of directional and positional offsets, curvature—a higher-order property of the trajectory—also exhibits a profound asymmetry. Specifically, the curvature of the inversion trajectory is substantially lower than that of the denoising trajectory.

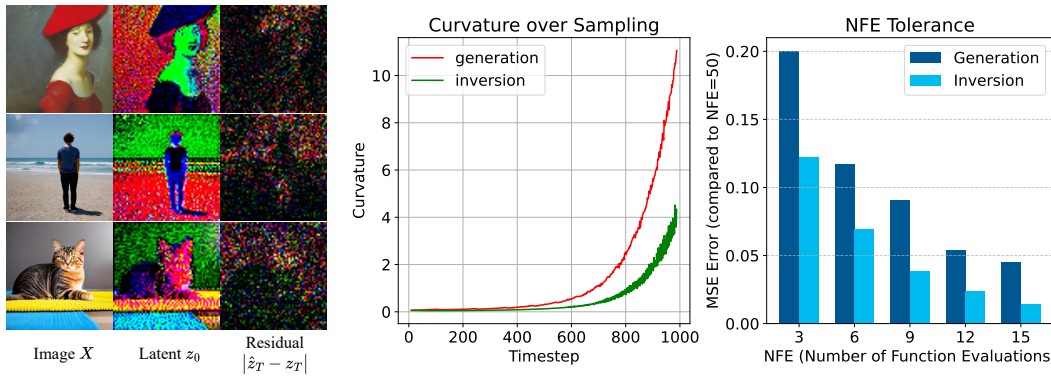

Figure 1: **Left**: Visualization of the inversion error, where latent vectors are projected down to 3 channels via PCA for display. **Middle**: Curvature of generation and inversion trajectories across diffusion timesteps. Discrete curvature estimated using 100 unconditionally generated images from Stable Diffusion v2.1. **Right**: The resulting error for generation and inversion when reducing the NFE, compared with a 50-step NFE baseline.

In Figure 1(middle), we illustrate the curvature differences between the 1000-step denoising (generation) and inversion trajectories. We observe that trajectories exhibit greater curvature near the noise end of the process (as $t \to T$). This is because the denoising network is trained on the forward diffusion process, where different images can diffuse to the same noise point, causing trajectory crossing (Lee et al., 2023). Consequently, in the early stages of denoising, the model must constantly correct its direction, leading to high curvature (Lee et al., 2023). This effect is particularly pronounced at the

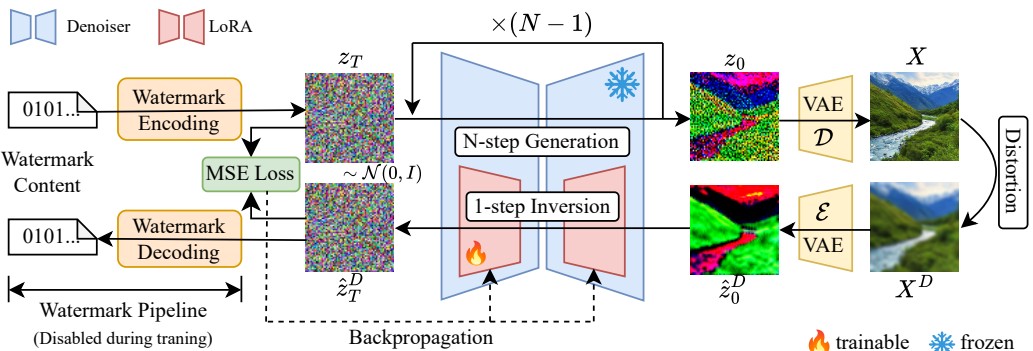

Figure 2: The framework of FARI. FARI simultaneously performs one-step distillation and adversarial training through a unified, end-to-end, LoRA-based fine-tuning process, enhancing both the efficiency and robustness of the inversion. The LoRA adapters are injected into the denoiser network and are deactivated during generation but activated for inversion. This strategy prevents any degradation of the original model's generation quality and eliminates the need to deploy a second, complete denoiser, making it highly memory-efficient.

very beginning when the latent variable is nearly pure noise. Once the fundamental semantics of the image have formed, the direction of progress becomes relatively fixed, and the trajectory's curvature decreases significantly. However, this high-curvature phenomenon is substantially less pronounced during the inversion process.

We attribute this partly to the fact that the accumulated error during inversion retains low-frequency information from the source image. As shown in Figure 1(left), partial outlines of the image remain visible in the noise reconstruction error, an observation consistent with prior work (Lin et al., 2024; Staniszewski et al., 2024; Nguyen et al., 2025). This residual semantic information helps to more accurately determine the correct direction of progress as the inversion approaches the noise end.

In general, a trajectory with lower curvature can be more accurately approximated with fewer linear steps (i.e., a lower NFE), as curvature is strongly correlated with the truncation error of the numerical solver. In the simple case where the curvature is zero, a single sampling step is sufficient. This key finding motivates us to explore the change in precision as the NFE is reduced for both generation and inversion. We decreased the NFE from 15 to 3, observing the deviation from the results of a standard 50-step NFE. The results in Figure 1(right) confirm that the inversion trajectory can indeed tolerate a much lower NFE, which provides the foundational premise for our proposed method by offering a significant increase in processing speed and, crucially, by enabling efficient adversarial training.

## 3.2 FARI

Guided by the geometric intuition that the inversion trajectory is highly compressible, our method is simple and effective. In essence, our strategy is to first find a low-NFE approximation of the DDIM inversion (Song et al., 2021) trajectory and then perform adversarial training upon this condensed path to achieve both speed and robustness. We use distillation, a common technique for accelerating diffusion models, to achieve the first step. While standard trajectory distillation (Salimans & Ho, 2022; Song et al., 2023; Yin et al., 2024a) for the generation process often requires days or even dozens of GPU-days and substantial memory, the favorable geometric properties of the DDIM inversion trajectory allow us to obtain a reasonably accurate low-NFE estimate with minimal effort. Although this initial estimate has some error (see Figure 3), we find this trade-off is acceptable in exchange for the immense gains in robustness and speed, and it has a negligible effect on the performance of the downstream watermarking tasks.

For simplicity and efficiency, we do not explicitly separate the distillation and adversarial fine-tuning into two distinct stages, as they share a consistent optimization objective and converge rapidly. It is crucial to note our departure from common distillation practices for the generation process. We

do not start with a real image and train our one-step model to mimic the output of a 50-step DDIM inversion. Instead, we sample a ground-truth Gaussian noise vector, perform the full generation process to obtain an image, and then learn a direct one-step mapping from this generated image back to the ground-truth initial noise. This approach avoids the performance ceiling imposed by the inherent inaccuracies of the 50-step DDIM inversion itself and is better aligned with the generative nature of the watermarking scenario.

Specifically, we fine-tune the denoising network of the diffusion model using Low-Rank Adaptation (LoRA) (Hu et al., 2022), a parameter-efficient fine-tuning technique that updates pretrained weight matrices through low-rank decomposition. Given a weight matrix $W_0 \in \mathbb{R}^{d \times k}$, the update is represented as $W_0 + \Delta W = W_0 + BA$, where $B \in \mathbb{R}^{d \times r}$, $A \in \mathbb{R}^{r \times k}$, and the rank $r \ll \min(d, k)$. During training, $W_0$ is frozen, and gradient updates are applied only to $A$ and $B$. The modified forward pass for an input $z$ becomes:

$$h = W_0 z + \Delta W z = W_0 z + BA z. \tag{8}$$

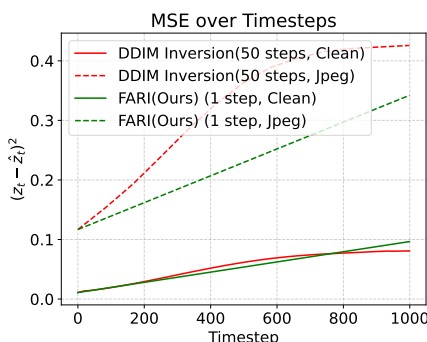

Figure 3: Inversion trajectory error of FARI and DDIM on clean and JPEG-compressed images.

By decomposing the full-rank matrix into the product of two low-rank matrices, LoRA significantly reduces the number of trainable parameters, thereby lowering memory usage.

As illustrated in Figure 2, each training loop proceeds as follows. We randomly sample an initial noise vector $z_T \sim \mathcal{N}(0, I)$ and a condition $c$ from a dataset $\mathcal{C}$ to generate an image $X$. During this generation phase, the LoRA branch is deactivated. The resulting image is then subjected to a distortion $D(\cdot)$ randomly selected from a predefined set $\mathcal{T}$, yielding a distorted image $X^D$, which is then encoded by the VAE encoder into a latent representation $z_0^D$. Subsequently, the LoRA branch is activated to perform a one-step inversion, which reconstructs the noise according to the following formula:

$$\hat{z}_T^D = \sqrt{\frac{\bar{\alpha}_T}{\bar{\alpha}_0}} z_0^D + \left( \sqrt{1 - \bar{\alpha}_T} - \sqrt{\frac{\bar{\alpha}_T(1 - \bar{\alpha}_0)}{\bar{\alpha}_0}} \right) \epsilon_\theta(z_0^D, 0, \emptyset; \psi). \tag{9}$$

Since $\bar{\alpha}_0 = 1$, this formula can also be equivalently written in the form of Eq.2:

$$\hat{z}_T^D = \sqrt{\bar{\alpha}_T} z_0^D + \sqrt{1 - \bar{\alpha}_T} \epsilon_\theta(z_0^D, 0, \emptyset; \psi), \tag{10}$$

where $\psi$ represents the LoRA parameters. For the inversion process, we use an unconditional setting (guidance scale = 1.0 and a null prompt). Prior works (Mokady et al., 2023; Wallace et al., 2023) have demonstrated that for standard DDIM inversion, an unconditional setting is often more precise because of the lack of invertibility in Classifier-Free Guidance (Ho & Salimans, 2022) (CFG). It is also important to note that we set the timestep $t = 0$ in the formula, rather than $t = T$ as expected from Eq. 7. This is because in a single-step scenario, the piecewise linear assumption is clearly violated. Empirically, we find that any other small timestep value ($t \approx 0$) can achieve performance comparable to $t = 0$, providing a much better match for the latent $z_0^D$, reducing the initial error and improving convergence. Finally, our training objective is defined as:

$$\min_{\psi} \mathbb{E}_{z_T, c \in \mathcal{C}, D \in \mathcal{T}} \left[ \| z_T - \hat{z}_T^D \|_2^2 \right] \tag{11}$$

Similarly, after training is complete, we deactivate the LoRA branch during denoising inference to preserve the original generation quality and enable it only for watermark extraction. This strategy is memory-efficient, eliminating the need to deploy two separate, largely identical denoisers. It is worth noting that the LoRA component can be regarded as a plug-and-play enhancement module. Even when it is removed, DDIM in principle allows inversion with arbitrary step counts, but the error may be very large. Further discussions, including details on the fine-tuned modules, training strategies, and hyperparameter selection, are provided in the ablation studies (Section4.4) and the Appendix E.3.

# 4 EXPERIMENTS

## 4.1 IMPLEMENTATION DETAILS

**Diffusion Models.** We selected Stable Diffusion v1.5 and v2.1 (Rombach et al., 2022) to cover the requirements of both the inversion baselines and the downstream watermarking task. For generation, we use a guidance scale of 7.5 and a number of function evaluations (NFE) of 50, employing the DDIM scheduler for all generations, except for those inversion methods that rely on their own specific sampling procedures.

**Watermarking Methods.** We conduct experiments with Tree-Ring (Wen et al., 2023) (TR) and Gaussian Shading (Yang et al., 2024) (GS), which embed watermarks in the frequency and spatial domains of the initial noise, respectively.

**Comparison Methods.** We evaluate our method against several categories of baselines for a comprehensive comparison. **First**, we establish standard benchmarks using 50-step DDIM inversion and one-step DDIM inversion. **Next**, we compare against methods specifically designed for high-fidelity inversion, including EDICT (Wallace et al., 2023) and BELM (Wang et al., 2024). We also consider ExactDPM (Hong et al., 2024), but owing to its extremely slow inference speed, we limit this comparison to the SD v2.1 model. **Finally**, given the scarcity of dedicated few-step inversion techniques, we adapt state-of-the-art acceleration methods originally designed for generation. For fast numerical solvers, we select AMED-Solver (Zhou et al., 2024a). Among the distillation-based methods, we include LCM-LoRA (Luo et al., 2023) and DMD2 (Yin et al., 2024a). Since the publicly available weights for these methods are limited to SD v1.5, these comparisons are performed only on that version. A detailed justification for our choice of baselines and a discussion of other related works are provided in the Appendix A.

**Training.** We train our model for 1,000 steps on 1,000 prompts from the MS-COCO-2017 dataset (Lin et al., 2014). The training is configured with a batch size of 4 and a learning rate of 1e-4. For LoRA, we use a rank of 8 and inject the adapters only into the attention-related modules. During the training loop, images are generated using an accelerated 20-step DDIM process to improve efficiency. Our adversarial distortion set includes 9 different augmentation types.

**Evaluation.** We evaluate all methods on a test set of 1,000 prompts from the Stable-Diffusion-Prompts (SDP) dataset[1] under a range of common distortions. To accurately assess practical performance, we bypass general image-level metrics and instead adopt the specific evaluation metrics defined by the downstream watermarking methods themselves. Specifically, for Gaussian Shading, we measure the bit-wise accuracy of the extracted message; for Tree-Ring, we measure the TPR at a stringent fixed FPR of $10^{-3}$, obtained by fitting the ROC curve on 1,000 positive and negative examples each and extrapolating.

All experiments are implemented in PyTorch 2.4.1 and run on a single NVIDIA RTX A6000 GPU. More details can be found in Appendix B.

## 4.2 MAIN RESULTS

The main results are presented in Table 1. Across both diffusion models and both downstream watermarking tasks, our FARI method achieves the most robust performance with the lowest NFE.

As anticipated, the performance of EDICT (Wallace et al., 2023) and BELM (Wang et al., 2024) on noise reconstruction is not as strong as their reported performance on image reconstruction. Their results are often inferior even to the standard DDIM baseline. This is particularly true for BELM; as a multi-step method, its errors appear to accumulate more rapidly, and it exhibits a significant sensitivity to mismatched guidance scales between the generation and inversion phases.

ExactDPM (Hong et al., 2024), which uses gradient descent to optimize the inversion trajectory, is extremely time-consuming but does show effectiveness against certain distortions like Gaussian noise. However, its objective is often misaligned with our task. For distortions involving missing content, such as random cropping or dropping, the method's attempts to inpaint the image can cause the trajectory to deviate in the wrong direction, harming noise reconstruction.

---

[1] https://huggingface.co/datasets/Gustavosta/Stable-Diffusion-Prompts

Table 1: Comparison of inversion methods on downstream watermarking methods under various image distortions.

| DM | Methods | NFE | Clean | Adv. | Jpeg | R.Crop | R.Drop | Resize | G.Blur | M.Blur | G.Noise | S&P | Bright |
|---|---|---|---|---|---|---|---|---|---|---|---|---|---|
| | | | **Bit Accuracy of Gaussian Shading Watermark** | | | | | | | | | | |
| SD v1.5 | DDIM | 50 | 0.9999 | 0.9777 | 0.9889 | **0.9781** | 0.9736 | 0.9975 | 0.9873 | 0.9983 | 0.9609 | 0.9354 | 0.9567 |
| | | 1 | 0.9999 | 0.9376 | 0.9703 | 0.8859 | 0.8808 | 0.9906 | 0.9585 | 0.9934 | 0.9398 | 0.9105 | 0.9085 |
| | EDICT | 50 | **1.0000** | 0.9637 | 0.9786 | 0.9656 | 0.9568 | 0.9969 | 0.9807 | 0.9985 | 0.9390 | 0.9124 | 0.9450 |
| | BELM | 50 | 0.9991 | 0.9465 | 0.9847 | 0.8960 | 0.8958 | 0.9939 | 0.9617 | 0.9956 | 0.9355 | 0.9275 | 0.9278 |
| | AMED[†] | 2 | **1.0000** | 0.9656 | 0.9807 | 0.9528 | 0.9462 | 0.9970 | 0.9808 | 0.9989 | 0.9587 | 0.9346 | 0.9410 |
| | LCM-LoRA | 2 | 0.9999 | 0.9541 | 0.9819 | 0.9352 | 0.9308 | 0.9924 | 0.9668 | 0.9955 | 0.9311 | 0.9030 | 0.9504 |
| | DMD2 | 1 | 0.9988 | 0.9287 | 0.9760 | 0.8446 | 0.8241 | 0.9792 | 0.9209 | 0.9836 | 0.9252 | 0.9007 | 0.9336 |
| | **FARI(Ours)** | **1** | **1.0000** | **0.9834** | **0.9935** | 0.9777 | **0.9761** | **0.9990** | **0.9957** | **0.9992** | **0.9836** | **0.9649** | **0.9612** |
| SD v2.1 | DDIM | 50 | **1.0000** | 0.9755 | 0.9892 | 0.9752 | 0.9669 | 0.9980 | 0.9860 | 0.9991 | 0.9590 | 0.9373 | 0.9447 |
| | | 1 | 0.9987 | 0.9359 | 0.9755 | 0.8841 | 0.8637 | 0.9748 | 0.9173 | 0.9819 | 0.9280 | 0.9069 | 0.9284 |
| | EDICT | 50 | **1.0000** | 0.9585 | 0.9773 | 0.9639 | 0.9558 | 0.9963 | 0.9805 | 0.9983 | 0.9396 | 0.9104 | 0.9395 |
| | BELM | 50 | 0.9990 | 0.9411 | 0.9847 | 0.8923 | 0.8938 | 0.9933 | 0.9602 | 0.9952 | 0.9333 | 0.9269 | 0.9334 |
| | AMED[†] | 2 | 1.0000 | 0.9662 | 0.9813 | 0.9559 | 0.9495 | 0.9970 | 0.9805 | 0.9989 | 0.9585 | 0.9357 | 0.9384 |
| | ExactDPM | > 150 | **1.0000** | 0.9670 | 0.9831 | 0.9675 | 0.9599 | 0.9974 | 0.9815 | 0.9987 | 0.9653 | 0.9241 | 0.9354 |
| | **FARI(Ours)** | **1** | **1.0000** | **0.9824** | **0.9941** | **0.9771** | **0.9700** | **0.9992** | **0.9956** | **0.9994** | **0.9815** | **0.9659** | **0.9588** |
| | | | **TPR@1e-3 of Tree-Ring Watermark** | | | | | | | | | | |
| SD v1.5 | DDIM | 50 | **1.000** | 0.949 | 0.989 | **1.000** | **1.000** | 0.999 | 0.996 | **1.000** | 0.636 | 0.946 | 0.972 |
| | | 1 | **1.000** | 0.863 | 0.905 | 0.602 | 0.649 | **1.000** | 0.994 | **1.000** | 0.891 | 0.990 | 0.737 |
| | EDICT | 50 | **1.000** | 0.942 | 0.975 | 0.998 | **1.000** | 0.999 | 0.992 | 0.997 | 0.605 | 0.954 | 0.962 |
| | BELM | 50 | 0.933 | 0.592 | 0.768 | 0.032 | 0.054 | 0.889 | 0.873 | 0.865 | 0.384 | 0.852 | 0.608 |
| | AMED[†] | 2 | **1.000** | 0.909 | 0.939 | 0.947 | 0.936 | 0.999 | 0.995 | 0.999 | 0.618 | 0.912 | 0.835 |
| | LCM-LoRA | 2 | **1.000** | 0.875 | 0.914 | 0.996 | 0.991 | 0.999 | 0.987 | 0.999 | 0.331 | 0.812 | 0.850 |
| | DMD2 | 1 | **1.000** | 0.760 | 0.709 | 0.116 | 0.473 | 0.996 | 0.971 | 0.995 | 0.913 | 0.985 | 0.678 |
| | **FARI(Ours)** | **1** | **1.000** | **0.997** | **1.000** | **1.000** | **1.000** | **1.000** | **1.000** | **1.000** | **0.980** | **1.000** | **0.992** |
| SD v2.1 | DDIM | 50 | **1.000** | 0.962 | 0.993 | **1.000** | **1.000** | **1.000** | 0.997 | **1.000** | 0.726 | 0.982 | 0.960 |
| | | 1 | **1.000** | 0.896 | 0.896 | 0.709 | 0.845 | **1.000** | 0.993 | 0.999 | 0.903 | 0.991 | 0.729 |
| | EDICT | 50 | **1.000** | 0.946 | 0.980 | 0.984 | 0.985 | 0.997 | 0.990 | 0.998 | 0.681 | 0.969 | 0.933 |
| | BELM | 50 | 0.882 | 0.543 | 0.721 | 0.001 | 0.000 | 0.803 | 0.787 | 0.801 | 0.417 | 0.812 | 0.541 |
| | AMED[†] | 2 | **1.000** | 0.926 | 0.966 | 0.957 | 0.983 | **1.000** | 0.998 | **1.000** | 0.656 | 0.983 | 0.795 |
| | ExactDPM | > 150 | **1.000** | 0.906 | 0.991 | 0.571 | 0.847 | **1.000** | 0.999 | **1.000** | 0.835 | 0.992 | 0.915 |
| | **FARI(Ours)** | **1** | **1.000** | **0.997** | **0.999** | **1.000** | **1.000** | **1.000** | **0.999** | **1.000** | **0.979** | **0.999** | **0.993** |

The acceleration methods also show limitations. The AMED-Solver[†] (Zhou et al., 2024a), which we fine-tuned adversarially in a similar manner to FARI, performs competitively. However, because its mechanism is limited to predicting a single median timestep, its solution space is too constrained to handle complex, real-world distortions, leaving a gap to the optimal performance. The distillation methods, LCM-LoRA (Luo et al., 2023) and DMD2 (Yin et al., 2024a), while highly effective for accelerating generation, do not transfer their success to the distinct task of one-step inversion. We posit that this is because predicting the reverse ODE direction from a highly structured image is a fundamentally different challenge than predicting it from pure noise.

## 4.3 GENERALIZATION

In this section, we explore FARI's generalization ability to different generation conditions. By default, we use the SD v2.1 model (Rombach et al., 2022) for these experiments and employ the standard 50-step DDIM inversion (Song et al., 2021) as the baseline. Experiments with different samplers and NFE in generation process can be found in Appendix E.

**Guidance Scales.** Given diverse user preferences for prompt adherence, higher guidance scales enforce the original prompt more strictly, while lower scales allow greater creative freedom. Our experiments span a wide range of 2.5 to 12.5. As shown in Figure 4(j), FARI's performance degrades only marginally under these settings.

**Noise Intensities.** To further test the robustness, we conduct experiments using different intensities of distortions. The results are shown in Figure 4(a-i). FARI consistently outperforms the standard DDIM baseline, and its advantage becomes even more pronounced as the distortion intensity increases.

**Noise types.** Although our model is trained with a rich set of augmentations, real-world distortions may include types unseen during training. To evaluate FARI's generalization ability against such unseen distortions, we compare the mean squared error (MSE) of the reconstructed noise under three

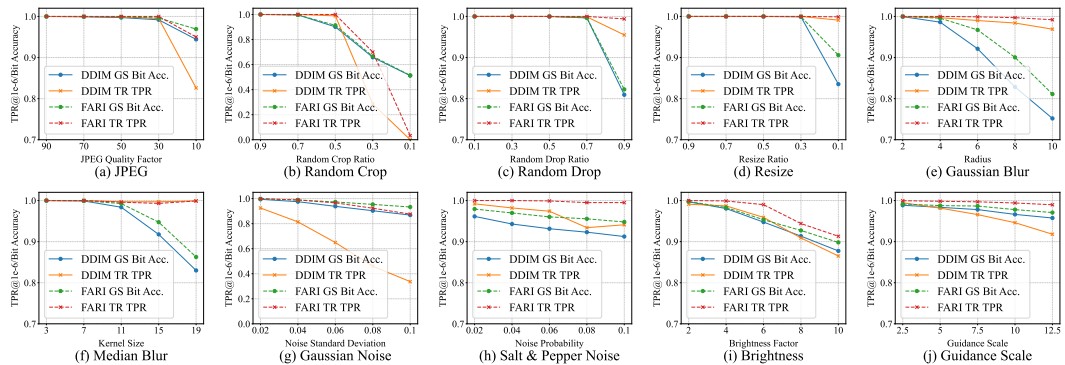

Figure 4: The results of experiments on various guidance scales and noise intensities.

distinct conditions: 1) **No Noise**, where the model is trained without any adversarial augmentations, serving as a baseline; 2) **Blind to Noise**, where FARI is trained on all distortions except for the one being tested; and 3) **All Noise**, our standard training procedure. The results, presented in Figure 5 (left), show that FARI achieves a notable improvement in robustness even against distortions it has never encountered during training.

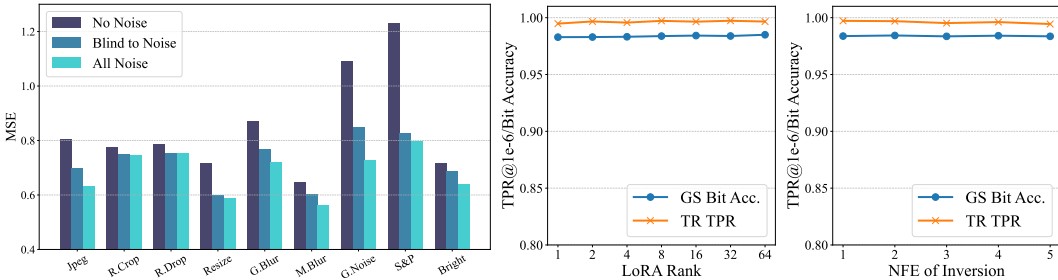

Figure 5: **Left**: Performance of FARI when trained under different noise settings, measured in mean squared error (MSE), where lower values are better. **Middle**: The effect of different LoRA ranks on FARI's performance. **Right**: The impact of using a higher NFE for end-to-end training on FARI's final performance.

## 4.4 ABLATION STUDY

In this section, we present ablation studies of FARI's training settings.

**LoRA Ranks.** We experimented with different LoRA ranks. Figure 5 (middle) shows that even with a rank of 1, the performance is considerable. Increasing the rank yields only marginal performance gains, confirming that a low rank is sufficient for our method.

**More NFE for Training.** We also attempted end-to-end training on a multi-step (NFE > 1) inversion. As shown in Figure 5 (right), using more steps not only sacrifices speed but also fails to improve performance, instead causing a slight degradation. We posit two reasons for this: first, a single step may already be sufficient to accurately approximate the low-curvature inversion trajectory; second, the non-trivial accumulation of errors during multiple forward passes may harm the final reconstruction. This confirms that our choice of a one-step inversion is optimal.

## 5 DISCUSSION

**Extended Application and Future Work.** We also briefly investigated FARI's performance on image reconstruction, given its nature as an inversion method. The specific results are available in the Appendix D. Furthermore, the reduction in NFE opens up another possibility. The adversarial removal of inversion-based watermarks has traditionally been difficult or extremely resource-intensive

due to the need for gradient propagation through the entire multi-step inversion process (Müller et al., 2024). Our one-step method may alleviate this, as it dramatically shrinks the computational graph. This will be a direction for our future work.

**Limitations.** Despite its strong performance in speed and robustness, FARI has two main limitations. First, its sacrificed precision on clean image inversions makes it unsuitable for direct application in image editing (Hertz et al., 2022). Second, like all inversion-based watermarks, FARI is dependent on ODE-based sampling (Song et al., 2021; Lu et al., 2022a) and will fail if an SDE sampler (Ho et al., 2020; Song et al., 2020) is employed.

## 6 CONCLUSION

Starting from the demands of watermark extraction, this paper identifies the critical bottleneck as the cumulative error arising from external distortions in the transmission channel. This leads us to question the necessity of traditional high-NFE inversion methods; they are not only slow and ineffective at mitigating these external errors but also computationally prohibit end-to-end adversarial training. Subsequently, we propose FARI, a framework built upon our key discovery of a geometric asymmetry: the inversion trajectory possesses a significantly lower curvature than its generation counterpart. This inherent compressibility allows us to efficiently distill the entire multi-step process into a single step. By doing so, FARI unlocks a robust adversarial training regime, creating an inverter that achieves state-of-the-art robustness and speed for practical, large-scale watermark verification.

### REPRODUCIBILITY STATEMENT

The resources required to reproduce the experiments of this paper are provided in the supplementary materials. This includes the complete source code for our proposed method, FARI, along with detailed instructions for setting up the environment and running the training and evaluation scripts. The implementation details for all baseline methods and experimental settings are described in Section 4.1 and Appendix B.

### ACKNOWLEDGMENTS

This work was supported in part by New Generation Artificial Intelligence-National Science and Technology Major Project (No. 2025ZD0123202) and by the National Natural Science Foundation of China under Grant 62472398, Grant U2336206, and Grant 62402469.

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

# A    DISCUSSION ABOUT RELATED WORKS

The body of work related to our method is extensive, primarily encompassing inversion techniques and diffusion model acceleration. We discuss these in separate categories below.

## A.1    INVERSION TECHNIQUES

A significant portion of inversion research focuses on improving the quality of image editing by enforcing trajectory symmetry. NTI (Mokady et al., 2023) first performs a standard DDIM inversion and then optimizes a null-text embedding for each step. This embedding sequence is then used during regeneration to ensure the new path closely follows the reverse of the inversion trajectory, achieving high-fidelity reconstruction. PTI (Dong et al., 2023) extends this by linking the embedding to the target prompt to enhance editing quality. Furthermore, NPI (Miyake et al., 2025) replaces the optimized null-text sequence with the prompt embedding itself, eliminating the need for optimization and greatly improving efficiency. Direct Inversion (Ju et al., 2023) also follows this path by decoupling the source and target diffusion branches. Notably, these methods do not actually change the result of the initial inversion; their focus is on the reconstruction phase. Therefore, their performance in a watermarking scenario is nearly identical to that of standard DDIM inversion, and we do not include them in our quantitative comparisons.

Another representative class of inversion methods, including AIDI (Pan et al., 2023), SPDInv (Li et al., 2024), ReNoise (Garibi et al., 2024), GNRI (Samuel et al., 2023), and ExactDPM (Hong et al., 2024), directly optimizes the inversion trajectory using fixed-point iteration or gradient descent. While these can indeed reduce the inversion error, they are often unstable. This instability is exacerbated by the initial value offset common in watermarking, which can cause the optimization to proceed in an incorrect direction or fail entirely. We found that while ExactDPM is the slowest, it is the most stable among them and also targets watermarking as a downstream task. Consequently, we select it as a representative of this category for comparison.

A third class of methods addresses the inherent lack of invertibility in the sampling process itself. Techniques like EDICT (Wallace et al., 2023), BDIA (Zhang et al., 2024a), and BELM (Wang et al., 2024) modify the sampler to achieve a smaller theoretical error bound. However, their design carries an implicit assumption that the entire process is free from external error. This holds true for reconstruction but is clearly violated in watermarking, where the image is subject to various distortions after generation that can push it out of the method's convergence domain. Furthermore, these methods can be more sensitive when handling images generated with a high guidance scale.

Recently, an interesting work in this area is SwiftEdit (Nguyen et al., 2025), which also achieves one-step inversion via model fine-tuning. However, it still considers the problem from an editing perspective, focusing on adapting to real images and enabling subsequent edits. While it achieves fast, high-quality results on editing tasks, it relies on base models that are already one-step generators (e.g., SwiftBrush v2 (Dao et al., 2024)) and has not demonstrated generalization to multi-step generators. This implies the trajectory it learns to fit is already a straight line, which is a simpler task. Additionally, its code and weights are not publicly available, precluding a direct comparison.

## A.2    DIFFUSION MODEL ACCELERATION

Given the lack of fast and stable inversion methods, we also include several diffusion model acceleration techniques in our baseline comparison. For the task of acceleration, we focus on methods that retain capabilities similar to the original model, rather than fully retraining a new one for faster generation. These methods can be divided into two categories. The first is based on higher-order solvers with lower truncation error, such as DPMSolver (Lu et al., 2022a). However, their performance ceiling is limited in extreme few-step scenarios. A noteworthy exception is the AMED-Solver (Zhou et al., 2024a), which is based on the mean value theorem for vector fields. It trains an additional small model to predict the median point of the trajectory, whose velocity can be used as the average velocity for the entire path, enabling sampling in as few as two steps.

The second category is distillation, where a student model learns to replicate the output of multiple teacher steps in a single step. This, however, often requires substantial training resources and time, which we argue is excessive if used solely for inversion. We select two representative baselines from

this rich field. LCM-LoRA (Luo et al., 2023) is similar to our method in that it stores the distilled parameters in a LoRA (Hu et al., 2022) module. However, its purpose is different: it is designed to adapt to various user-personalized models, enabling accelerated sampling without requiring a separate distillation for each fine-tuned model. DMD2 (Yin et al., 2024a), on the other hand, uses distribution matching distillation. It does not directly learn the teacher's output but rather its target distribution, allowing the student's performance to surpass the ceiling of the original teacher model. Another relevant work is SFD (Zhou et al., 2024b), which observes the smooth modification of the gradient field and, inspired by this, focuses training resources on essential timesteps. While known for its fast training speed for a distillation method, it is still slower to train than FARI and its one-step performance is inferior to our chosen baseline, DMD2. Our results show that distillation methods do not perform exceptionally well on the inversion task. As we posited, while the prediction target is similar, the input is fundamentally different: predicting a direction from a highly structured image is a distinct challenge compared with predicting it from pure noise.

## B    DETAILED EXPERIMENTS SETTINGS

In this section, we provide the specific details of our experimental setup.

**Models and Generation.** We conduct experiments on Stable Diffusion (SD) v1.5 and v2.1 (Rombach et al., 2022). The generation process utilizes the `diffusers` library in Python, with pretrained weights sourced from the Hugging Face Hub repositories `runwayml/stable-diffusion-v1-5` and `stabilityai/stable-diffusion-2-1-base`, respectively. All images are generated at a resolution of $512 \times 512$ pixels. For testing, we use 50-step DDIM sampling with a fixed guidance scale of 7.5, which are common settings for the downstream watermarking tasks.

**Watermarking Methods.** For both the Tree-Ring (Wen et al., 2023) and Gaussian Shading (Yang et al., 2024) watermarks, we use the official open-source code provided by the authors on GitHub. We use Tree-Ring in its `rand` mode, with the watermark embedded in the fourth channel of the latent space. For Gaussian Shading, we adopt the default settings ($f_{ch} = 4, f_h = 8, f_w = 8$, for a 256-bit capacity) and use a stream cipher to encrypt the watermark message.

**Baselines.** For DDIM (Song et al., 2021), we use our own implementation, which is equivalent to using the `DDIMInverseScheduler` from `diffusers`. It is critical to note that the `timestep_spacing` must be set to `"trailing"` instead of the default `"leading"`. The `"leading"` setting introduces a significant error at low NFEs because it sets the first `prev_timestep` to 0. Since the scheduler calculates the current `timestep` by subtracting a step size from `prev_timestep`, this can result in a negative timestep, which is then clipped to 0. This effectively makes the first step a non-operation (from $t = 0$ to $t = 0$). The `"trailing"` setting avoids this error. While both settings yield similar results at high NFEs, the difference is substantial at very few steps, especially when NFE=1. Furthermore, our one-step DDIM inversion baseline, similar to FARI, uses $t = 0$ when predicting epsilon, as using $t = T$ introduces a large error.

For EDICT (Wallace et al., 2023) and BELM (Wang et al., 2024), we use the implementation provided by BELM. The NFE is set to 50 steps, and EDICT's hyperparameter $p$ is set to 0.93 as recommended in the original paper. For ExactDPM (Hong et al., 2024), we use the official implementation with the Backward Euler method for inversion. To ensure a fair and efficient comparison, we disable the additional decoder inversion. We set its inversion step count to 10, but note that it requires an iterative optimization process that often takes over 150 iterations, depending on the distortion type and intensity.

As the official weights for AMED-Solver (Zhou et al., 2024a) were not provided for our target models, we retrained it ourselves. Crucially, we performed an end-to-end adversarial training for the inversion task, similar to our FARI training, and aligned its parameter count with FARI. Since it requires a minimum of two steps for sampling, we set its NFE to 2. For LCM-LoRA (Luo et al., 2023) and DMD2 (Yin et al., 2024a), we use the officially provided weights. We set LCM-LoRA to use two steps (within its allowed 2-8 range) and use a single step for DMD2, which has one-step sampling capabilities.

**Training and Evaluation.** For FARI's training, we use a batch size of 4 with the Adam optimizer and a learning rate of 1e-4. When retraining the AMED-Solver from scratch, we adjusted the learning rate to 1e-3. For our training dataset, we use prompts associated with the MS-COCO-2017 dataset (Lin et al., 2014); while the original dataset lacks prompts, we utilize the captions provided in the official Tree-Ring repository. For testing, we use the test split of the Stable-Diffusion-Prompts (SDP) dataset.

**Distortions.** The types of distortions used in our training and testing are illustrated in Figure 6. Geometric distortions, such as rotation and scaling, were excluded, as we consider their handling to be more related to the intrinsic robustness mechanism of the watermark itself rather than the inversion process.

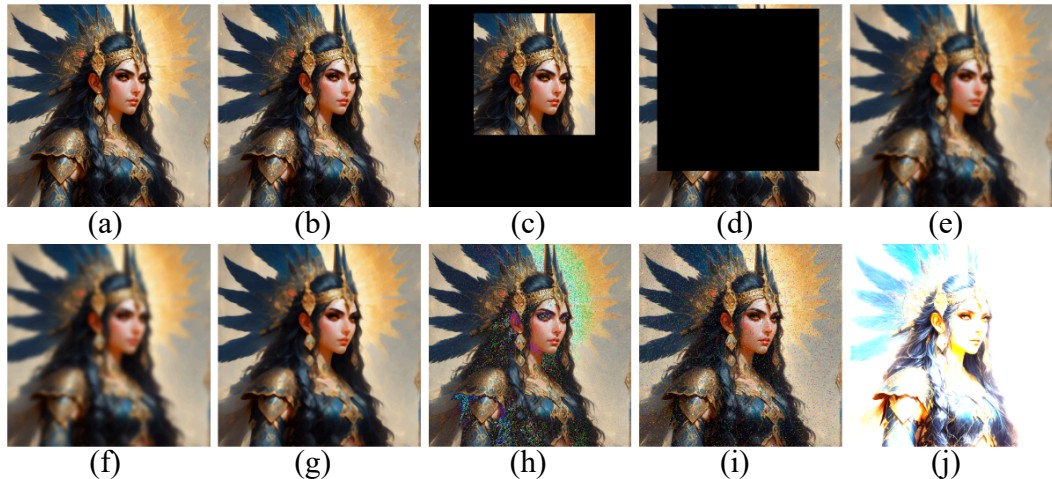

Figure 6: Visualization of the distortion set used in our experiments. (a) Clean image or identity transformation. (b) JPEG, $QF = 25$. (c) 60% area Random Crop (R.Crop). (d) 80% area Random Drop (R.Drop). (e) 25% Resize and restore (Resize). (f) Gaussian Blur, $r = 4$ (G.Blur). (g) Median Blur, $k = 7$ (M.Blur). (h) Gaussian Noise, $\mu = 0$, $\sigma = 0.05$ (G.Noise). (i) Salt and Pepper Noise, $p = 0.05$ (S&P). (j) Brightness, $factor = 6$ (Bright).

## C  DETAILS ABOUT THE CURVATURE EVALUATION.

In Section 3.1, we measure the curvature of the generation and inversion trajectories as a function of the timestep. The experiment is conducted on the Stable Diffusion v2.1 model (Rombach et al., 2022), generating 100 images. Both the generation and inversion processes are set to be unconditional to eliminate interference from other factors. The curvature is approximated using a formal definition of discrete curvature.

Specifically, for a trajectory of latent variables $\{x_t\}_{t=0}^T$, we consider three consecutive points $x_{t+1}$, $x_t$, and $x_{t-1}$. We first define the two corresponding velocity vectors as $v_t = x_{t-1} - x_t$ and $v_{t+1} = x_t - x_{t+1}$. We then approximate the arc length $s_t$ as:

$$s_t = \|v_{t+1}\|_2 + \|v_t\|_2 \tag{12}$$

The discrete curvature $\kappa_t$ at timestep $t$ is then defined as the angle $\theta_t$ between the velocity vectors, divided by the arc length $s_t$:

$$\kappa_t = \frac{\theta_t}{s_t} \tag{13}$$

where the angle $\theta_t$ is given by:

$$\theta_t = \arccos\left(\frac{v_{t+1} \cdot v_t}{\|v_{t+1}\|_2 \|v_t\|_2}\right) \tag{14}$$

We also conducted two additional experiments, with the results shown in the Figure 7. For conditional generation, we use prompts from the Stable-Diffusion-Prompts dataset. All other settings remain identical to our main experiments. First, we used conditional generation to create an image, and then compared the curvature of the original generation trajectory against two types of inversion: one that was conditionally-aligned and one that was unconditional. The results show that the curvature of both inversion trajectories is consistently lower than that of the generation trajectory. It is important to note, however, that lower curvature does not necessarily equate to higher inversion accuracy. Second, we measured the curvature of a regeneration trajectory (from the inverted noise). The results indicate that the regenerated path also has a lower curvature than the initial generation path. This further validates our conclusion that the residual low-frequency semantic information in the reconstructed noise partly helps to reduce trajectory curvature.

To strengthen the generalizability of our findings, we conduct additional on the COCO dataset and using Stable Diffusion v3.5 Medium (SD v3.5M) (Esser et al., 2024). The results are presented in the Figure 8.

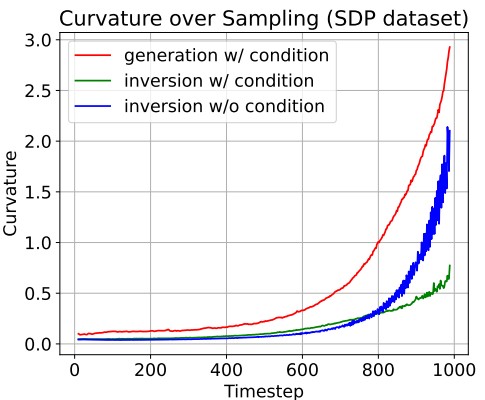 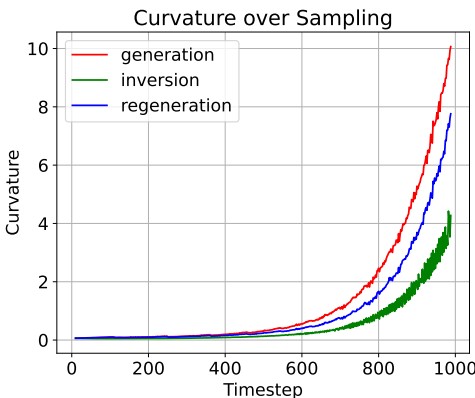

Figure 7: **Left**: The curvature of the conditional generation, unconditional inversion, and conditional inversion trajectories over the diffusion timesteps. **Right**: The curvature of the generation, inversion, and regeneration trajectories over the diffusion timesteps.

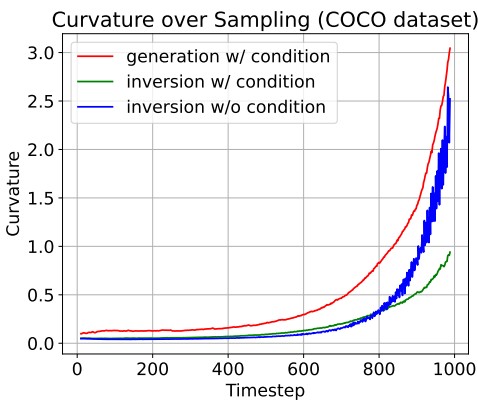 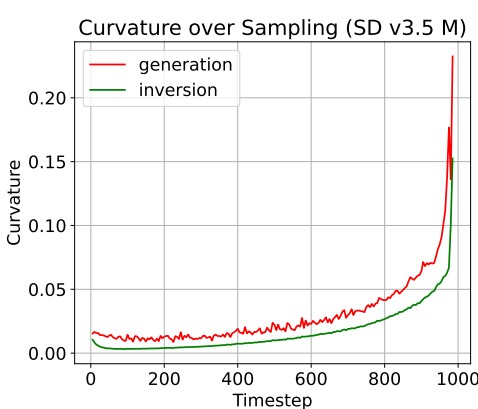

Figure 8: **Left**: The curvature of the the the conditional generation, unconditional inversion, and conditional inversion trajectories over the diffusion timesteps, measured using SD v2.1 on the COCO dataset . **Right**: The curvature of the generation and inversion trajectories over the diffusion timesteps, measured using SD v3.5M.

# D    EXTENDED APPLICATION ON IMAGE RECONSTRUCTION AND EDITING

Although FARI is specifically designed for watermark extraction, its remarkable speed prompted us to investigate its potential for image reconstruction and editing. We therefore conducted an exper-

iment comparing the peak signal-to-noise ratio (PSNR) of images reconstructed by FARI against those reconstructed by DDIM (Song et al., 2021), EDICT (Wallace et al., 2023), BELM (Wang et al., 2024), and ExactDPM (Hong et al., 2024). The results are presented in Table 2. While disabling adversarial training (FARI$^{Cln}$) boosts FARI's precision on clean inversions, a gap remains when compared to leading editing-oriented methods. Regarding image editing, many techniques require attention map or feature sharing during the intermediate steps of the inversion and regeneration process to ensure high consistency. As our one-step method lacks these intermediate steps for intervention, it is not directly compatible with mainstream plug-and-play diffusion-based editing frameworks (Hertz et al., 2022). We hope our work can serve as inspiration for subsequent research in this area.

Table 2: The performance of inversion methods on image reconstruction.

|  | DDIM | EDICT | BELM | ExactDPM | FARI | FARI$^{Cln}$ |
|---|---|---|---|---|---|---|
| PSNR | 17.33 | 25.26 | 24.04 | 20.02 | 10.61 | 14.60 |

# E    MORE EXPERIMENTAL RESULTS

## E.1    DISCUSSION ABOUT TRAINING TIME AND INFERENCE TIME

**Training Time**   Figure 9 illustrates the training dynamics of FARI, showing the exponentially weighted moving average (EWMA) of the loss curve alongside the performance on the Gaussian Shading (Yang et al., 2024) and Tree-Ring (Wen et al., 2023) watermark tasks. The entire training process, including the online construction of data pairs, takes approximately 70 minutes on a single NVIDIA RTX A6000 GPU. Notably, FARI surpasses the performance of the 50-step DDIM baseline on both watermarking tasks around the 300th training step (approximately 20 minutes), highlighting the efficiency of our method.

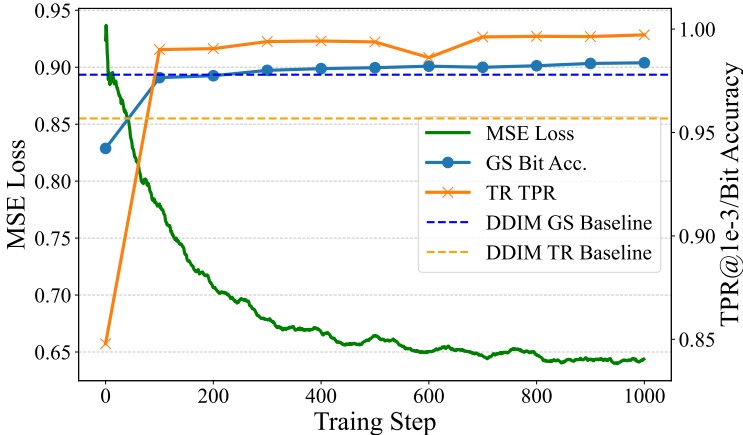

Figure 9: The loss tendency and tested performance on downstream watermarking tasks over the training step.

**Inference Time**   The use of a LoRA (Hu et al., 2022) modules affects the denoiser's inference efficiency, with the average time per function evaluation on an NVIDIA RTX A6000 GPU increasing from 0.0247 s to 0.0358 s. While this introduces latency, the total time remains less than that of a two-step inversion. To eliminate this overhead during inference, a practical solution is to merge the LoRA weights by $W' = W_0 + \Delta W = W_0 + BA$. This can be achieved by creating an extra fused copy of the LoRA-injected module, allowing the system to switch between the original module for image generation and the fine-tuned, robust module for one-step watermark inversion. Alternatively, for systems with sufficient VRAM, a simpler but more memory-intensive approach is to load an entirely separate model with the weights pre-merged.

### E.2 MORE RESULTS ABOUT GENERALIZATION

**Samplers.**     Users may employ samplers different from the one used during training. To test FARI's generalization to this possibility, we conducted sampling using a variety of common ODE solvers (Song et al., 2021; Zhao et al., 2023; Liu et al., 2022; Zhang & Chen, 2022; Lu et al., 2022a). The results, shown in Table 3, indicate that FARI outperforms the DDIM inversion baseline across all tested samplers.

Table 3: Generalization to different samplers. For each cell, the values represent TR TPR / GS Bit Acc.

|       | DDIM
(Song et al., 2021) | UniPC
(Zhao et al., 2023) | PNDM
(Liu et al., 2022) | DEIS
(Zhang & Chen, 2022) | DPMSolver
(Lu et al., 2022a) |
|-------|---------------|---------------|---------------|---------------|---------------|
| DDIM  | 0.966 / 0.9780 | 0.886 / 0.9550 | 0.967 / 0.9757 | 0.963 / 0.9744 | 0.884 / 0.9549 |
| FARI  | 0.997 / 0.9841 | 0.951 / 0.9637 | 0.997 / 0.9846 | 0.997 / 0.9826 | 0.951 / 0.9664 |

**Inference Steps of Generation.**     To optimize the training speed, we used a fixed 20 steps for image generation during our training process. While this may be a low number of inference steps for the selected models, we present results with a greater number of generation steps in Table 4. The results demonstrate that FARI, despite being trained under these efficient conditions, generalizes well to higher-quality generation settings and consistently outperforms the baseline.

Table 4: Generalization to different NFEs of generation. For each cell, the values represent TR TPR / GS Bit Acc.

|       | 25 | 50 | 100 |
|-------|-----|-----|------|
| DDIM  | 0.965 / 0.9752 | 0.966 / 0.9780 | 0.964 / 0.9768 |
| FARI  | 0.997 / 0.9826 | 0.997 / 0.9841 | 0.997 / 0.9843 |

**Other Models.**     We evaluate FARI's performance on SD v3.5M (Esser et al., 2024) and SDXL-Turbo (Podell et al., 2023). Since SD v3.5M does not support DDIM sampler, we implement inversion for this model via naive Euler Method. SDXL-Turbo inherently supports one-step generation, so we compare the performance of one-step DDIM inversion against FARI.

Given that SD v3.5 has 16 latent channels, we set $f_{ch} = 2$ and $f_h = f_w = 8$ for Gaussian Shading, resulting in a capacity of 512 bits. For Tree-Ring Watermark, we use only the last four channels for watermark embedding. The corresponding results are shown in Table 5 and Table 6, demonstrating FARI's strong generalization capability. FARI exhibits substantial improvements on both downstream watermarking approaches, especially for Tree-Ring. In the presence of distortions, naive inversion fails to maintain the viability of Tree-Ring watermarking, demonstrating that FARI significantly broadens the practical applicability of inversion-based watermarking techniques.

Table 5: Performance of naive inversion and FARI on SD v3.5M.

| Methods | NFE | Clean | Adv. | JPEG | R.Crop | R.Drop | Resize | G.Blur | M.Blur | G.Noise | S&P | Bright |
|---------|-----|-------|------|------|--------|--------|--------|--------|--------|---------|-----|--------|
| **Bit Accuracy of Gaussian Shading Watermark** | | | | | | | | | | | | |
| Naive | 20 | 0.9994 | 0.9178 | 0.9440 | 0.9654 | 0.9403 | 0.9437 | 0.8728 | 0.9609 | 0.7778 | 0.8736 | 0.9818 |
| FARI | 1 | 0.9995 | 0.9515 | 0.9671 | 0.9713 | 0.9469 | 0.9756 | 0.9384 | 0.9852 | 0.8765 | 0.9191 | 0.9833 |
| **TPR@1e-3 of Tree-Ring Watermark** | | | | | | | | | | | | |
| Naive | 20 | 0.958 | 0.161 | 0.020 | 0.364 | 0.543 | 0.000 | 0.000 | 0.001 | 0.009 | 0.003 | 0.512 |
| FARI | 1 | 0.998 | 0.978 | 0.994 | 0.998 | 0.995 | 0.995 | 0.952 | 0.996 | 0.917 | 0.968 | 0.985 |

**More Datasets.**     To further validate generalization across datasets, we additionally select 1,000 prompts from a new dataset (DiffusionDB[2]). Combined with the existing SDP and COCO datasets, we conduct cross-dataset validation. The results are shown in Table 7, presented in the format of TR TPR / GS Bit Acc., with 50-step DDIM as the baseline.

---

[2]https://huggingface.co/datasets/poloclub/diffusiondb

Table 6: Performance of naive inversion and FARI on SDXL-Turbo.

| Methods | NFE | Clean | Adv. | JPEG | R.Crop | R.Drop | Resize | G.Blur | M.Blur | G.Noise | S&P | Bright |
|---------|-----|-------|------|------|--------|--------|--------|--------|--------|---------|-----|--------|
| **Bit Accuracy of Gaussian Shading Watermark** | | | | | | | | | | | | |
| DDIM | 1 | 0.9755 | 0.8576 | 0.9142 | 0.7449 | 0.7552 | 0.9370 | 0.8841 | 0.9346 | 0.8935 | 0.8229 | 0.8315 |
| FARI | 1 | 0.9998 | 0.9511 | 0.9903 | 0.9115 | 0.9161 | 0.9871 | 0.9370 | 0.9922 | 0.9801 | 0.9354 | 0.9102 |
| **TPR@1e-3 of Tree-Ring Watermark** | | | | | | | | | | | | |
| DDIM | 1 | 0.561 | 0.201 | 0.672 | 0.000 | 0.000 | 0.065 | 0.001 | 0.221 | 0.412 | 0.271 | 0.163 |
| FARI | 1 | 0.998 | 0.855 | 0.967 | 0.743 | 0.933 | 0.893 | 0.683 | 0.928 | 0.949 | 0.837 | 0.763 |

Table 7: Cross-dataset generalization results. Performance is reported as TR TPR / GS Bit Acc.

| Training Dataset \ Test Dataset | SDP | COCO | DiffusionDB |
|---------------------------------|-----|------|-------------|
| DDIM Baseline (50-step) | 0.966 / 0.9780 | 0.964 / 0.9785 | 0.962 / 0.9763 |
| SDP | — | 0.997 / 0.9852 | 0.995 / 0.9839 |
| COCO | 0.997 / 0.9841 | — | 0.997 / 0.9846 |
| DiffusionDB | 0.995 / 0.9832 | 0.997 / 0.9861 | — |

**Removal Attacks.** To evaluate FARI's robustness against watermark removal attacks, we test three regeneration attacks and one adversarial optimization attack. We evaluate three different regeneration attacks. For Zhao et al. (2024), we set the noise steps to 300. For Cheng et al. (2020) and Ballé et al. (2018), we set the quality parameter to 3 (representing the highest attack strength). For the adversarial optimization attack (Lukas et al., 2023), we use the following settings: $\epsilon = 4/255$, Adam optimizer, learning rate of 0.01, and 5 optimization steps per image, all of which are consistent with the default configuration. Note that the attack implementation does not involve Gaussian Shading (GS). Since the GS verification process is non-differentiable and relies on sign-based verification, we optimized by minimizing the MSE distance between the reconstructed noise after optimization and the negative of the original noise. This can cause more elements in the reconstructed noise to flip their signs and proved to be effective (otherwise, the bitwise accuracy will be 1.0000, which means the attack is useless). The corresponding results are shown in Figure 10. FARI provides improvements in robustness. However, the ability to resist attacks remains largely dependent on the underlying design of the base watermarking method, as FARI serves as a plug-and-play inversion approach to enhance their detection performance.

### E.3 MORE ABLATION STUDIES

**Trained Modules.** We experimented with training different modules of the denoiser. By default, we only fine-tune the linear layers within the attention-related modules. In an *Extended LoRA FT* setting, we fine-tune all linear and convolutional layers using LoRA. We also tested full fine-tuning as an upper bound. The results are shown in Table 8. Although training more modules yields a marginal improvement, the increase is not significant. We therefore opted for the more parameter-efficient default setting.

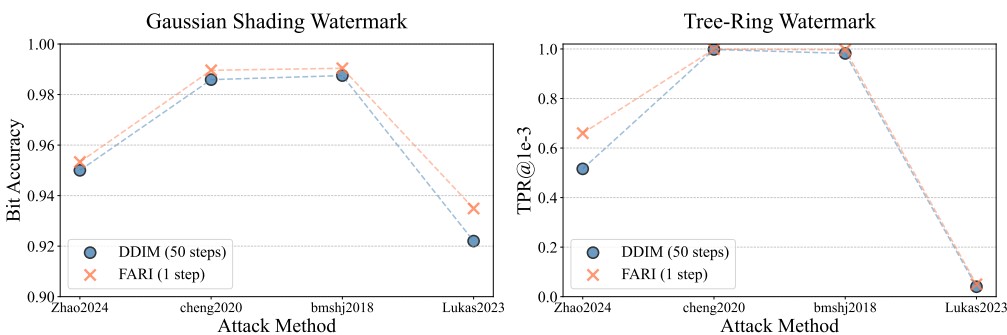

Figure 10: Performance of DDIM inversion and FARI under different attacks.

**Training Strategies.** We compared several training strategies, including a non-adversarial baseline. We also tested *regular distillation*, where the model learns to mimic the output of a 50-step DDIM inversion on real images, as well as a step-wise adversarial tuning similar to diffusion pre-training (*regular fine-tuning*) (Ho et al., 2020). For *regular distillation*, the performance ceiling effect is obvious if we were to directly learn from DDIM's inversion of a distorted image. Therefore, our actual learning objective is defined as the L2 distance between the noise produced by the student model on an augmented image and the noise produced by the baseline DDIM on the corresponding clean image. For *regular fine-tuning*, the evaluation is conducted using a 50-step inversion. The results in Table 9 confirm our claims: adversarial training is vital for FARI's robustness, the *regular distillation* approach is limited by its teacher's performance ceiling, and the step-wise objective of *regular fine-tuning* fails to learn global robustness.

Table 8: The Performance of FARI on different trained modules.

|  | TR TPR | GS Bit Acc. |
| --- | --- | --- |
| Attention LoRA FT | 0.997 | 0.9841 |
| Extended LoRA FT | 0.996 | 0.9849 |
| Full FT | 0.998 | 0.9860 |

Table 9: The Performance of FARI on different training strategies.

|  | TR TPR | GS Bit Acc. |
| --- | --- | --- |
| FARI w/ adversarial training | 0.997 | 0.9841 |
| FARI w/o adversarial training | 0.919 | 0.9788 |
| Regular distillation | 0.995 | 0.9823 |
| Regular fine-tuning | 0.963 | 0.9774 |

**Choice of $t$.** Regarding the choice of $t$ value in Equation 9, we conducted an ablation study by training with different $t$ values. The results are shown in Table 10. The results demonstrate that $t \approx 0$ serves as a better initialization for optimization than larger $t$ values.

Table 10: Ablation study on different $t$ values in Equation 9.

|  | $t = 0$ | $t = 100$ | $t = 500$ | $t = 999$ |
| --- | --- | --- | --- | --- |
| GS Bit Acc. | 0.9841 | 0.9835 | 0.9809 | 0.9772 |
| TR TPR | 0.997 | 0.996 | 0.995 | 0.988 |

## E.4 General Evaluation and Visual Results

To demonstrate FARI's broader compatibility with inversion-based watermarking methods beyond Gaussian Shading (Yang et al., 2024) and Tree-Ring (Wen et al., 2023), we directly measure and report the MSE of FARI's reconstructed latent noise in Table 11, with visual results shown in Figure 11.

Table 11: The MSE of inversion methods under various image distortions.

| DM | Methods | NFE | Clean | Adv. | Jpeg | R.Crop | R.Drop | Resize | G.Blur | M.Blur | G.Noise | S&P | Bright |
| --- | --- | --- | --- | --- | --- | --- | --- | --- | --- | --- | --- | --- | --- |
| SD v1.5 | DDIM | 50 | **0.2003** | 0.9285 | 0.8740 | 0.8734 | 0.8834 | 0.7671 | 0.9706 | 0.6920 | 1.1860 | 1.2963 | 0.8139 |
|  |  | 1 | 0.7108 | 0.9016 | 0.8981 | 0.9646 | 0.9621 | 0.8294 | 0.8987 | 0.8157 | 0.9067 | 0.9316 | 0.9074 |
|  | EDICT | 50 | 0.2646 | 1.0477 | 1.0400 | 1.0407 | 1.0547 | 0.8569 | 1.0756 | 0.7654 | 1.2852 | 1.3735 | 0.9370 |
|  | BELM | 50 | 0.6593 | 1.1723 | 1.0230 | 1.4621 | 1.4384 | 0.9181 | 1.1744 | 0.8789 | 1.2193 | 1.2065 | 1.2305 |
|  | AMED[†] | 2 | 0.3577 | 0.8375 | 0.8022 | 0.8756 | 0.8800 | 0.7116 | 0.8733 | 0.6439 | 0.9562 | 1.0033 | 0.7917 |
|  | LCM-LoRA | 2 | 0.4488 | 0.8471 | 0.7773 | 0.8264 | 0.8331 | 0.7296 | 0.8614 | 0.7126 | 1.0413 | 1.0904 | 0.7517 |
|  | DMD2 | 1 | 0.7934 | 0.9201 | 0.9478 | 0.9472 | 0.9392 | 0.8673 | 0.8999 | 0.8578 | 0.9367 | 0.9587 | 0.9266 |
|  | **FARI(Ours)** | 1 | 0.2033 | **0.6699** | **0.6138** | **0.7385** | **0.7396** | **0.5650** | **0.7037** | **0.5486** | **0.7146** | **0.7920** | **0.6135** |
| SD v2.1 | DDIM | 50 | **0.2051** | 0.9657 | 0.9070 | 0.9395 | 0.9614 | 0.8012 | 1.0167 | 0.7080 | 1.1861 | 1.2946 | 0.8769 |
|  |  | 1 | 0.6314 | 0.9573 | 0.8634 | 0.9934 | 1.0049 | 0.8861 | 1.0091 | 0.8471 | 1.0543 | 1.0456 | 0.9116 |
|  | EDICT | 50 | 0.2659 | 1.0458 | 1.0419 | 1.0383 | 1.0476 | 0.8560 | 1.0745 | 0.7646 | 1.2700 | 1.3691 | 0.9503 |
|  | BELM | 50 | 0.6706 | 1.1705 | 1.0213 | 1.4556 | 1.4313 | 0.9263 | 1.1767 | 0.8872 | 1.2099 | 1.1954 | 1.2190 |
|  | AMED[†] | 2 | 0.3538 | 0.8357 | 0.8025 | 0.8729 | 0.8791 | 0.7095 | 0.8726 | 0.6424 | 0.9522 | 0.9990 | 0.7915 |
|  | ExactDPM | > 150 | 0.2495 | 1.1306 | 1.0733 | 1.2565 | 1.2645 | 0.9105 | 1.1577 | 0.8165 | 1.2495 | 1.4110 | 1.0359 |
|  | **FARI(Ours)** | 1 | 0.2214 | **0.6862** | **0.6335** | **0.7461** | **0.7530** | **0.5905** | **0.7216** | **0.5638** | **0.7294** | **0.7962** | **0.6418** |

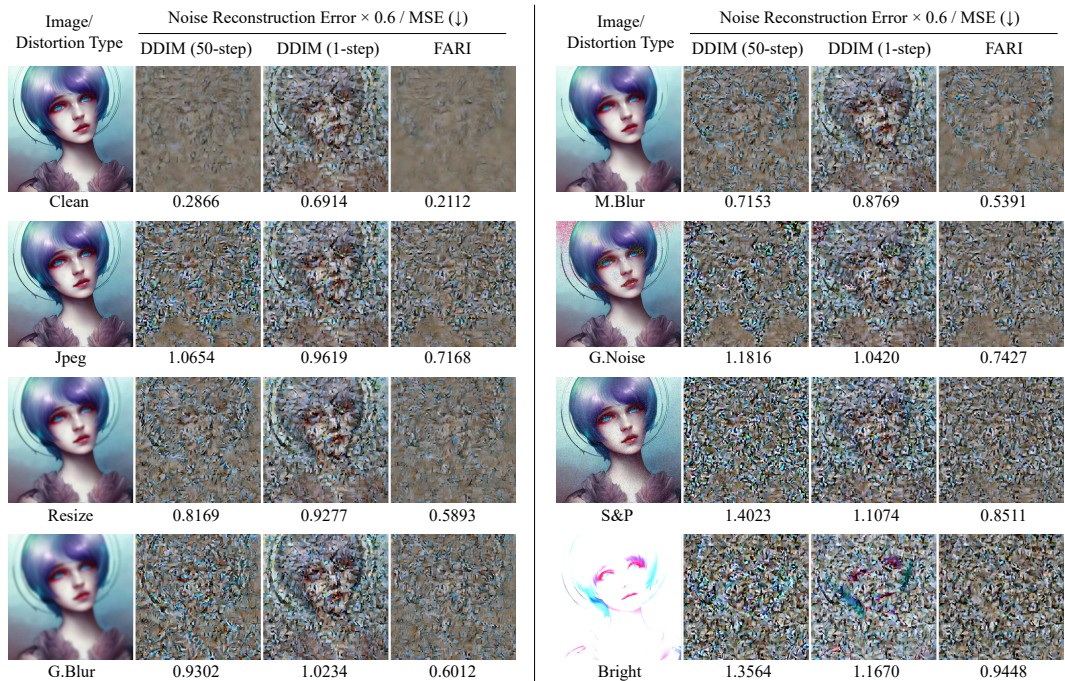

| Image/Distortion Type | Noise Reconstruction Error × 0.6 / MSE (↓) | | | Image/Distortion Type | Noise Reconstruction Error × 0.6 / MSE (↓) | | |
|---|---|---|---|---|---|---|---|
| | DDIM (50-step) | DDIM (1-step) | FARI | | DDIM (50-step) | DDIM (1-step) | FARI |
| Clean | 0.2866 | 0.6914 | 0.2112 | M.Blur | 0.7153 | 0.8769 | 0.5391 |
| Jpeg | 1.0654 | 0.9619 | 0.7168 | G.Noise | 1.1816 | 1.0420 | 0.7427 |
| Resize | 0.8169 | 0.9277 | 0.5893 | S&P | 1.4023 | 1.1074 | 0.8511 |
| G.Blur | 0.9302 | 1.0234 | 0.6012 | Bright | 1.3564 | 1.1670 | 0.9448 |

Figure 11: FARI reduces reconstruction error, especially under distortion. Naive DDIM single-step inversion produces visible artifacts. Error scaled by 0.6 to avoid oversaturation.

## F  LLM USAGE STATEMENT

A large language model (LLM) was used as an assistive tool for the writing and editing of this paper. Its primary function was to polish and rephrase sentences for improved clarity, readability, and academic style. The authors have reviewed and take full responsibility for all content presented.

