# OpenReview forum: "FARI: Robust One-Step Inversion for Watermarking in Diffusion Models"
_ICLR.cc/2026/Conference — ICLR 2026 Poster_

### Official Review · Reviewer_yTFb · 2025-10-15

**Soundness:** 2
**Presentation:** 3
**Contribution:** 3
**Rating:** 4
**Confidence:** 4

**Summary:**

This paper presents a method for one-step robust inversion in the context of diffusion model-generated images. This method leverages LoRA fine-tuning with the insight that the inversion trajectory has lower curvature than the forward step, making it easier to reproduce with fewer steps. They specifically speed up the inversion process with minimal loss of quality for watermark extraction, demonstrating that their method outperforms the existing baseline of DDIM in both speed and quality when used with the Tree-Ring and Gaussian Shading watermarks. They also show that this improvement is maintained in adversarial settings where the images have been perturbed naturally. Finally, they study the robustness of their method to varying attack and guidance parameters and perform an ablation study over some of their hyperparameters.

**Strengths:**

While the process itself is not particularly original, the curvature observation appears novel. The use case of faster inversion for watermarking and how it enables better adversarial training is significant.
The results showcase a tangible improvement over the existing baseline and other adapted methods.
The paper is well-written.

**Weaknesses:**

1. While the paper is well-written, it suffers from a lack of clarity around key points, such as the exact purpose of the LoRA branch, and it glosses over some important substitutions in the equations.

2. The experimentation could benefit from an evaluation of FARI against stronger watermark removal methods [1,2], rather than only natural distortions.

3. The experiments use the metric of TPR at $10^6$ FPR, but they only evaluate with $10^3$ samples. This appears to be a mismatch, either the FPR is extrapolated or the sample size needs further description. Likewise, they are using four significant digits, even though the measurement is based on an evaluation with only 1000 samples. This would be dependent on how many bits were used per-key, but the number is not specified. Was each prompt sample used with different watermarking keys repeatedly? If so, how many? Either the FPR is meaningless, or the exact number of samples used needs clarification. These issues cast a shadow of doubt on the credibility of your results. The exact experimentation and evaluation setup needs to be clarified to justify that your results are meaningful.

[1] Zhao, Xuandong, et al. "Invisible image watermarks are provably removable using generative ai." Advances in neural information processing systems 37 (2024): 8643-8672.

[2] Lukas, Nils, et al. "Leveraging optimization for adaptive attacks on image watermarks." arXiv preprint arXiv:2309.16952 (2023).

**Questions:**

1. It should be clarified whether the LoRA branch is purely for robustness or is necessary for the one-step inversion process as a whole. What this means is, if we remove the LoRA parameters and their associated fine-tuning from Equation (10), does the inversion process still work? If it does not work, clarifying that LoRA is necessary for the entire process, rather than just for robustness, could be valuable.

2. For Equations (9) and (10), I have a few questions and concerns:

2.a. It is specified that $t$ is set to 0 instead of $t=T$ as one would normally do for inversion when following the piecewise linearity assumption. Rightfully, you highlight that for a one-step process, the piecewise linearity assumption does not hold, but that does not really justify your choice of $t=0$ rather than any other $t$ value, like $t=T$ or $t=1$. Also, from my understanding, $t=0$ is never used when training the denoising network (since $t=0$ is the last step), so it would be out-of-distribution. Also, you then mention that empirically, you use a small $t \sim 0$ value, but never specify what it is, how it is chosen, and why it is not just zero as you specified in Equations (9) and (10). Perhaps an ablation study over different values of $t$ could help clarify the choice.

2.b. You mention that the piecewise linearity assumption is broken, and you mention that Equation (10) is equivalent to Equation (2). Yet, in Equation (2), the $\epsilon$ component is dependent on $z_t$, but for Equation (10), you substituted it with $z_0$. This would only be possible if the piecewise linearity assumption held, allowing you to substitute with minimal error. This substitution is not explained nor justified. My intuition is that the LoRA training helps mitigate the error caused by this substitution.

3. Please clarify, either in the text or the figure caption in Figure (1), both what the unit of measurement is for curvature and how this curvature was measured. Has this been computed over one sample, one hundred samples, 1000 samples? Since this is your motivating result, it should be clear how it was derived. Showing curvature statistics across multiple seeds, model architectures, and datasets could help solidify your observation as a general one.

4. This is minor, but the figure text can sometimes be quite small to read, especially for small figures like Figure 3.

---

> ### Author Response · Authors · 2025-11-19
> **Rebuttal by Authors (1/2)**
>
> Thank you for your insightful comments on our work. Our responses to each of the points you raised are detailed below.
>
> ---
>
> ## **Re: Role of LoRA (W1 & Q1)**
> The LoRA module stores all parameter updates during distillation and adversarial fine-tuning, so it contributes to both speed and robustness. The LoRA module is disabled during generation and enabled during inversion (watermark extraction) to avoid the memory overhead of loading two SD models simultaneously, as we mention in the paper. If the LoRA component is removed, DDIM in principle allows inversion with arbitrary step counts, but the error may be very large. Please refer to our quantitative evaluation (Table 1 and 7) and visual results (Figure 10) for details.
>
> ---
>
> ## **Re: Regarding Equations 9 and 10 (W1 & Q2)**
>
> Thank you for your detailed and highly professional questions regarding Equations 9 and 10.
>
> Equation 9 can be directly viewed as the DDIM inversion formula. Without any substitution, its form should be:
>
> $\boldsymbol{z}_T = \sqrt{\frac{\bar{\alpha}\_T}{\bar{\alpha}\_{0}}} \boldsymbol{z}\_{0} + \left( \sqrt{1-\bar{\alpha}\_T} - \sqrt{\frac{\bar{\alpha}\_T(1 - \bar{\alpha}\_{0})}{\bar{\alpha}\_{0}}} \right) \boldsymbol{\epsilon}\_\theta(\boldsymbol{z}\_T, T).$
>
> However, as stated in Section 3.1, we cannot explicitly compute $\boldsymbol{\epsilon}\_\theta(\boldsymbol{z}\_T, T)$on the right side. Typically, we would use the piecewise linear assumption for substitution. However, we point out that this assumption does not hold over such a large span from $T$ to 0. Nevertheless, we still perform the substitution, which motivates your question (Q2.b). Our rationale is actually quite simple: as FARI is an inversion method with NFE=1, it can only use $\boldsymbol{z}_0$ as input—this choice is not based on error considerations. Your intuition is correct: error correction is indeed handled by LoRA training.
>
> We then explain why we do not use $t=T$. This is because the piecewise linear assumption fails, and using $T$ does not help us better approximate $\boldsymbol{\epsilon}\_\theta(\boldsymbol{z}_T, T)$ with $\boldsymbol{\epsilon}\_\theta(\boldsymbol{z}\_0, T)$. Instead, replacing $T$ with $0$ makes $\boldsymbol{\epsilon}\_\theta(\boldsymbol{z}\_0, 0)$ a better optimization starting point. Adjusting the timestep schedule during distillation is a common practice.
>
> In the paper, we did not claim that Equation 10 is equivalent to Equation 2. The original statement is: "this formula (Eq. 9) can also be equivalently written in the form of Eq. 2: Eq. 10," meaning Equation 9 is equivalent to Equation 10, and Equation 10 shares the same simple form as Equation 2.
>
> We use $t=0$ because by convention, we denote the latent obtained by directly encoding an image through the VAE as $\boldsymbol{z}_0$. Our empirical tests show that it performs similarly to other values close to 0. For consistency, we choose $t=0$. Since we set $t=0$ in the formula, we also use 0 in practice.
>
> Regarding the concern that $t=0$ is typically out-of-distribution (OOD), you are correct. To avoid issues related to this OOD setting, we added clarification that $t \approx 0$ yields similar functionality, not just $t=0$. However, that statement may not have conveyed our intention accurately. We have now revised it and other potentially ambiguous passages.
>
> Following your suggestion, we also provide ablation experiments on $t$. The results show that $t\approx 0$ serves as a better initialization for optimization than $t=T$.
>
> |  | t=0 | t=100 | t=500 | t=999 |
> | --- | --- | --- | --- | --- |
> | GS Bit Acc | 0.9841 | 0.9835 | 0.9809 | 0.9772 |
> | TR TPR | 0.994 | 0.993 | 0.991 | 0.986 |

---

> > ### Comment · Reviewer_yTFb · 2025-11-19
> > **Response to rebuttal**
> >
> > I thank the authors for their timely response.
> >
> > # LoRA
> > My original comment was towards the lack of clarity in the paper. I eventually understood its purpose, but clarifying that purpose in the paper could help readers understand why LoRA is necessary.
> >
> > # Re: Equations
> > I understand that you only have $z_0$ for inversion since you are doing it in one step, and therefore you have to use it. I believe that the ablation study over $t$ should be included in the paper to help justify your decision.

---

> ### Author Response · Authors · 2025-11-19
> **Rebuttal by Authors (2/2)**
>
> ## **Re: Robustness against various attacks (W2)**
>
> We evaluate three different regeneration attacks. For [1], we set the noise steps to 300. For [2] and [3], we set the quality parameter to 3 (representing the highest attack strength). The corresponding results are as follows:
>
> GS Bit. Acc.:
>
> | Attack | DDIM (50 steps) | FARI (1 step) |
> | --- | --- | --- |
> | diff_attacker_300 [1] | 0.9500 | 0.9533 |
> | cheng2020-anchor_3 [2] | 0.9859 | 0.9896 |
> | bmshj2018-factorized_3 [3] | 0.9875 | 0.9904 |
>
> TR TPR:
>
> | Attack | DDIM (50 steps) | FARI (1 step) |
> | --- | --- | --- |
> | diff_attacker_300 [1] | 0.516 | 0.660 |
> | cheng2020-anchor_3 [2] | 0.998 | 0.999 |
> | bmshj2018-factorized_3 [3] | 0.982 | 0.997 |
>
> The results show that FARI provides improvements in robustness. However, the ability to resist attacks remains largely dependent on the underlying design of the base watermarking method, as FARI serves as a plug-and-play inversion approach to enhance their detection performance.
>
> Regarding the second attack you mentioned [4], which relies on optimization, we were unable to find an open-source implementation and found it difficult to reproduce its optimization process for inversion-based watermarks in our environment. However, we believe that FARI, or indeed any method designed to enhance watermark robustness, would likely struggle to help these watermarking methods resist such attacks, as this may be related to fundamental security considerations.
>
> [1] Invisible Image Watermarks Are Provably Removable Using Generative AI, NeurIPS 2024
>
> [2] Learned Image Compression with Discretized Gaussian Mixture Likelihoods and Attention Modules, CVPR 2020
>
> [3] Variational image compression with a scale hyperprior, ICLR 2018
>
> [4] Leveraging Optimization for Adaptive Attacks on Image Watermarks, ICLR 2024
>
> ---
>
> ## **Re: Computation of TPR@1e-6 (W3)**
>
> Gaussian Shading is a multi-bit watermark that embeds 256 bits of information by default, with random content and keys for each prompt. We report its bit-wise accuracy, retaining four decimal places. Tree-Ring is a single-bit watermark, for which we report TPR@FPR, retaining three decimal places.
>
> For Tree-Ring, we indeed use the extrapolation method. Each evaluation uses 1,000 unwatermarked images and 1,000 watermarked images to fit the ROC curve and extrapolate to the target FPR. This follows the same measurement methodology for Tree-Ring used in published papers [5][6]. We have added the corresponding description to the revised paper.
>
> [5] Gaussian Shading: Provable Performance-Lossless Image Watermarking for Diffusion Models，CVPR 2024
>
> [6] ROAR: Reducing Inversion Error in Generative Image Watermarking, ICCV 2025
>
> ---
>
> ## **Re: Computation of curvature (Q3)**
> Curvature is computed over 100 samples. Its unit is the reciprocal of length, but since length is generally unitless in this context, we did not explicitly specify units. We provide detailed explanation of the computation methodology in Appendix C. Following your suggestion, we have now moved portions of this explanation to the figure caption. We have also supplemented experiments across additional model architectures and datasets to strengthen the generalizability of our findings.
> Additionally, in the original figure, curvature values were incorrectly scaled by a factor of 4. This has now been corrected.
>
> ---
>
> ## **Re: Figure text (Q4)**
> We have made targeted adjustments to improve readability, especially for Figure 3.

---

> ### Comment · Reviewer_yTFb · 2025-11-19
> **Second response to rebuttal**
>
> # W2
> Thank you for these additional experiments and results; they are valuable in showing the quality of your work and could be worth including in the paper. As for [4], I found the following public implementation by the authors: https://github.com/nilslukas/adaptive-watermark-attacks
>
> # FPR
> Thank you for clarifying regarding the decimal places. I, however, still take issue with the TPR rate selected and the extrapolation. Extrapolating from a FPR of $1e-3$ to $1e-6$ can be very misleading, especially given the lack of theoretical justification for why such an extrapolation would hold for your method and the fact that you only have 1000 unwatermarked samples (which is 1000x smaller than the smallest set you would need to make such a claim without extrapolation). It is not uncommon for models/ML methods to behave differently at extremely low FPRs; therefore, extrapolation may be inaccurate or even misleading. Also, the paper you referenced [5] is an even worse offender in this regard, claiming a FPR of $10^-10$, i.e., one in ten billion, from only 1000 samples. They do not justify why, and you did not provide additional supporting evidence as to why it is fine to extrapolate from $1e-3$ to $1e-6$ for the TPR. In the Tree-Ring watermark paper [6], they use the same setup as you do, 1000 unwatermarked images and 1000 watermarked images, but they only evaluate at 1\% FPR. I recommend you report the numbers at a $1e-3$ FPR, or follow the Tree-Ring watermark paper and use a 1\% FPR.
> Please update the FPR for your experiments to reflect the number of samples you are working with, or provide a theoretical proof that supports that the Tree-Ring watermark supports such extrapolation of the FPR.
>
> [4] Leveraging Optimization for Adaptive Attacks on Image Watermarks, ICLR 2024
>
> [5] Gaussian Shading: Provable Performance-Lossless Image Watermarking for Diffusion Models， CVPR 2024
>
> [6] Wen, Y., Kirchenbauer, J., Geiping, J., & Goldstein, T. (2023). Tree-ring watermarks: Fingerprints for diffusion images that are invisible and robust. arXiv preprint arXiv:2305.20030.

---

> > ### Author Response · Authors · 2025-11-20
> >
> > Thank you for your quick response!
> >
> > ---
> > ## **Regarding LoRA**
> >
> > Regarding the LoRA approach, we appreciate your feedback and have added clarification about its purpose and application in Section 3.2 to address your concern.
> >
> > ---
> > ## **About additional experimental results**
> >
> > We will incorporate the ablation study on $t$ and the performance of FARI under various attacks into the paper. But to maintain consistency between the figure numbering in the paper and the references in our rebuttal (for all reviewers), we are temporarily adding only textual modifications.
> >
> > Thank you for providing the attack implementation. We conducted experiments using the code with the following settings: ε = 4/255, Adam optimizer, learning rate of 0.01, and 5 optimization steps per image, all of which are consistent with the default configuration. Note that the attack implementation does not involve Gaussian Shading (GS). Since the GS verification process is non-differentiable and relies on sign-based verification, we optimized by minimizing the MSE distance between the reconstructed noise after optimization and the negative of the original noise. This can cause more elements in the reconstructed noise to flip their signs and proved to be effective (otherwise, the bitwise accuracy will be 1.0000, which means the attack is useless).
> >
> > The corresponding results (TR TPR@1% / GS Bit. Acc.) are shown below:
> >
> > |  | DDIM (50 steps) | FARI (1 step) |
> > | --- | --- | --- |
> > | adaptive watermark attack | 0.04 / 0.9220 | 0.05 / 0.9349 |
> >
> > FARI effectively improves the attack robustness of GS, while the improvement for TR remains limited. As we mentioned, when the base watermark method itself struggles to resist a certain type of attack, FARI alone is insufficient to achieve significant improvements in robustness against that attack. Such a clarification will also be include in the paper as well.
> >
> > ---
> >
> > ## **Regarding FPR**
> >
> > Thank you for your suggestion. We have recalculated TPR@1e-3 based on the previous results (logs or intermediate results) and revised the corresponding sections in the paper accordingly. Figures that need to be modified have also be redrawn.
> >
> > We also present the results below:
> >
> > SD v1.5:
> >
> > | Method | NFE | Clean | Adv. | JPEG | R.Crop | R.Drop | Resize | G.Blur | M.Blur | G.Noise | S&P | Bright |
> > | --- | --- | --- | --- | --- | --- | --- | --- | --- | --- | --- | --- | --- |
> > | DDIM | 50 | 1.000 | 0.949 | 0.989 | 1.000 | 1.000 | 0.999 | 0.996 | 1.000 | 0.636 | 0.946 | 0.972 |
> > | DDIM | 1 | 1.000 | 0.863 | 0.905 | 0.602 | 0.649 | 1.000 | 0.994 | 1.000 | 0.891 | 0.990 | 0.737 |
> > | EDICT | 50 | 1.000 | 0.942 | 0.975 | 0.998 | 1.000 | 0.999 | 0.992 | 0.997 | 0.605 | 0.954 | 0.962 |
> > | BELM | 50 | 0.933 | 0.592 | 0.768 | 0.032 | 0.054 | 0.889 | 0.873 | 0.865 | 0.384 | 0.852 | 0.608 |
> > | AMED | 2 | 1.000 | 0.909 | 0.939 | 0.947 | 0.936 | 0.999 | 0.995 | 0.999 | 0.618 | 0.912 | 0.835 |
> > | LCM-LoRA | 2 | 1.000 | 0.875 | 0.914 | 0.996 | 0.991 | 0.999 | 0.987 | 0.999 | 0.331 | 0.812 | 0.850 |
> > | DMD2 | 1 | 1.000 | 0.760 | 0.709 | 0.116 | 0.473 | 0.996 | 0.971 | 0.995 | 0.913 | 0.985 | 0.678 |
> > | FARI | 1 | 1.000 | 0.997 | 1.000 | 1.000 | 1.000 | 1.000 | 1.000 | 1.000 | 0.980 | 1.000 | 0.992 |
> >
> > SD v2.1:
> >
> > | Method | NFE | Clean | Adv. | JPEG | R.Crop | R.Drop | Resize | G.Blur | M.Blur | G.Noise | S&P | Bright |
> > | --- | --- | --- | --- | --- | --- | --- | --- | --- | --- | --- | --- | --- |
> > | DDIM 50 | 50 | 1.000 | 0.962 | 0.993 | 1.000 | 1.000 | 1.000 | 0.997 | 1.000 | 0.726 | 0.982 | 0.960 |
> > | DDIM 1 | 1 | 1.000 | 0.896 | 0.896 | 0.709 | 0.845 | 1.000 | 0.993 | 0.999 | 0.903 | 0.991 | 0.729 |
> > | EDICT | 50 | 1.000 | 0.946 | 0.980 | 0.984 | 0.985 | 0.997 | 0.990 | 0.998 | 0.681 | 0.969 | 0.933 |
> > | BELM | 50 | 0.882 | 0.543 | 0.721 | 0.001 | 0.000 | 0.803 | 0.787 | 0.801 | 0.417 | 0.812 | 0.541 |
> > | AMED | 2 | 1.000 | 0.926 | 0.966 | 0.957 | 0.983 | 1.000 | 0.998 | 1.000 | 0.656 | 0.983 | 0.795 |
> > | ExactDPM | >150 | 1.000 | 0.906 | 0.991 | 0.571 | 0.847 | 1.000 | 0.999 | 1.000 | 0.835 | 0.992 | 0.915 |
> > | FARI | 1 | 1.000 | 0.997 | 0.999 | 1.000 | 1.000 | 1.000 | 0.999 | 1.000 | 0.979 | 0.999 | 0.993 |

---

> > > ### Comment · Reviewer_yTFb · 2025-11-27
> > > **Response**
> > >
> > > Thank you for the quick response and running the experiments so quickly.
> > >
> > > Given that you have addressed my comments and concerns, I am increasing my score.

---

> > > > ### Author Response · Authors · 2025-11-27
> > > >
> > > > Thank you for your response and for your valuable recognition of our work! Your suggestions are helpful to us, and we commit to incorporating the additional experimental results mentioned in our rebuttal.

---

### Official Review · Reviewer_3yip · 2025-10-23

**Soundness:** 3
**Presentation:** 3
**Contribution:** 2
**Rating:** 6
**Confidence:** 3

**Summary:**

This paper proposes a solution to the bottleneck of inversion-based watermarking in diffusion models: the diffusion inversion process is slow, unreliable, and particularly lacks robustness against external distortions. The authors' core observation is that the curvature of the DDIM inversion trajectory is significantly lower than its forward generation trajectory, making the inversion path highly compressible and suitable for approximation with very few NFEs. Based on this, the paper proposes a one-step inversion framework named FARI. The method achieves this one-step inversion through lightweight adversarial LORA fine-tuning, which unlocks end-to-end robustness training. Experiments show that with only about 20 minutes of fine-tuning, FARI's one-step inversion surpasses the robustness of the 50-step DDIM inversion baseline while dramatically increasing speed.

**Strengths:**

1. Novel Insight about Trajectory Curvature: The paper's main contribution is its core observation—that the DDIM inversion trajectory's curvature is significantly lower than the generation trajectory. The authors correctly identify this geometric property as a key "enabler," showing that the inversion path is inherently highly compressible and thus well-suited for one-step approximation via techniques like distillation. This is a non-obvious and highly original insight.

2. Practical Implementation: The use of lightweight LoRA for fine-tuning is a very practical and elegant solution. It requires minimal training time (i.e., ~20 minutes) and allows the "plug-in" robustness enhancements to be applied during the inversion, without affecting the original model's weights, thus perfectly preserving generation quality. This effective design is highly practical for real-world usage.

**Weaknesses:**

Weakness

1. Technical Novelty: While the geometric insight is novel, the methods used (e.g., trajectory distillation, adversarial training, LoRA fine-tuning) are all established techniques. The paper's contribution lies more in the clever application and combination of these existing tools rather than a fundamental breakthrough in inversion or distillation methodology itself.

2. Reliance on Unconditional Inversion: The FARI inverter is trained and executed in an unconditional setting (i.e., guidance scale = 1.0) to avoid the irreversibility issues of CFG. However, most images are generated using a much larger CFG. This mismatch between guided generation and unguided inversion is a potential source of error and a limitation on the method's generality, even if adversarial training empirically compensates for it.

3. Limited Applicability: The method and the inversion-based watermarks it relies on are fundamentally tied to ODE-based deterministic samplers (e.g., DDIM). The authors acknowledge this in the limitations section. It is incompatible with SDE samplers, which limits its universality in the broader diffusion model applications.

**Questions:**

Questions

1.  Regarding unconditional inversion (refer to the Weakness): If the inversion is performed unconditionally (CFG=1.0) while generation uses strong guidance (e.g., CFG=7.5). How does this unguided inversion robustly map back to the correct $z_T$? Does the adversarial training implicitly learn to "undo" the CFG effect, and have the authors explored alternatives?

2.  Regarding inversion curvature: The paper's core claim is the lower curvature of the inversion trajectory, which is attributed to "low-frequency information retained in the accumulated error." This explanation feels more like an intuitive observation than a formal analysis. Could the authors provide a more in-depth, formal theoretical analysis explaining why the inversion trajectory systematically exhibits lower curvature?

---

> ### Author Response · Authors · 2025-11-19
> **Rebuttal by Authors**
>
> Thank you for your helpful feedback. We provide our responses to each point below.
>
> ---
>
> ## **Re: Concerns about novelty (W1)**
> The primary goal of this paper is to simultaneously improve the extraction efficiency and robustness of inversion-based watermarks through a novel inversion method—a pressing problem that urgently needs to be addressed. To this end, we analyze the limitations of existing inversion methods and identify that they fail to account for external distortions, rendering their acceleration or accuracy improvement mechanisms ineffective in practice.
> As mentioned in the introduction, adversarial training is a natural approach. However, the challenge lies in the fact that multi-step inversion incurs prohibitively high backpropagation costs, while few-step inversion may compromise accuracy. This naturally leads us toward distillation techniques. While this direction is conceptually straightforward, the critical insight is that current widely-used high-precision trajectory distillation techniques for diffusion models require extremely high training costs, making distillation an impractical solution for accelerating watermark extraction. This is where our curvature-based discovery plays a crucial role: it provides the foundation for fast and effective distillation, enabling us to propose the FARI method.
> You can see our reasoning pathway for addressing this problem—our insights are self-consistent and tightly integrated with our method.
>
> ---
>
> ## **Re: FARI's generalization across CFG settings (W2 & Q1)**
>
> We measure FARI's generalization across various CFG settings in our experiments. During training, we only use CFG=7.5 for generation. However, in Sec 4.3 (Fig 4(j)), we evaluate CFG values ranging from 2.5 to 12.5. FARI demonstrates better generalization than the DDIM baseline (degrading more slowly), suggesting that FARI indeed learns to mitigate the impact of CFG (though not eliminate it entirely).
>
> FARI functions more as a general robust enhancement paradigm. For more targeted solutions to CFG's impact on inversion, we believe modifying the underlying design may be more effective. For instance, CFG++[1] provides analysis from a manifold perspective and offers a simple yet effective solution that may be helpful for your purposes.
>
> [1] CFG++: Manifold-constrained Classifier Free Guidance For Diffusion Models, ICLR 2025.
>
> ---
>
> ## **Re: Regarding SDE (W3)**
> We acknowledge this limitation and aim to address it in future work. However, given that ODE-based models currently represent the mainstream, FARI remains effective for the majority of existing diffusion models.
>
> ---
>
> ## **Re: Theoretical explanation of curvature findings (Q2)**
>
> As described in Sec 3.1, inversion performs a substitution: $\boldsymbol{\epsilon}\_\theta(\boldsymbol{z}\_t, t) \approx \boldsymbol{\epsilon}\_\theta(\boldsymbol{z}\_{t-1}, t)$. Excluding the low-frequency information effects in accumulated error that we have mentioned, we believe this substitution critically impacts curvature, including changes in its value and its time derivative. However, analyzing these changes involves examining derivatives of black-box neural network functions, which also requires experimental validation. Given that systematic analysis aims to derive more generalizable conclusions, if our theoretical analysis still relies on experimental studies across different models, directly measuring and presenting curvature results appears more intuitive.
>
> To enhance the generalizability of this finding, we validate it across additional model architectures and datasets in the Appendix C.

---

> > ### Comment · Reviewer_3yip · 2025-11-25
> >
> > I appreciate the authors' prompt response to my comments. The additional details you provided are helpful and certainly offer more context for your work.
> >
> > However, given the inherent limitations of the method and its relative novelty, I believe it is appropriate to uphold my original score.

---

> > > ### Author Response · Authors · 2025-11-25
> > >
> > > Thank you for your prompt response and for maintaining your positive evaluation of our work. We are pleased to address your questions.

---

### Official Review · Reviewer_8NsM · 2025-10-30

**Soundness:** 2
**Presentation:** 2
**Contribution:** 2
**Rating:** 2
**Confidence:** 5

**Summary:**

This paper implements a learning-based inversion for Diffusion Models using LoRA, and names the method FARI. Also, by using the fact that the inversion trajectory has a small curvature, it reduces NFE to 1. This allows performing inversion with very low runtime after a single training. The paper applies this to the Tree-ring watermark, a famous application of diffusion inversion, and achieves faster inference time and good accuracy compared to many baselines.

**Strengths:**

1. This work seems quite original, as this tries learning-based inversion of diffusion models, and such approches are seldom found in the existing works. (But I have a concern for this; which will be discussed in Weakness)

2. Table 1 shows that the proposed method works well on Tree-Ring Watermarks (Wen et al., 2023).

**Weaknesses:**

1. The proposed method uses a learning-based inversion with LoRA. I personally do not think learning-based inversion methods basically give good fidelity. This is because of the basic information difference between calculating an image from (image + noise) and calculating (image + noise) from an image. This is why many cited inversion papers use mathematical methods instead of training a new one (Wallace et al., 2023; Pan et al., 2023; Hong et al., 2024; and so on). Again, I basically think the paper's approach makes accurate noise estimation difficult. However, in the main experiment in Section 4.2, the current Tree-Ring watermark might be just too strong. This could be why FARI showed good watermark detection performance (TPR and bit accuracy) in Table 1 (I also suspect the experiment is just too easy, because all other baselines also have very high bit accuracy). But, reconstructing the noise $z_T$ well is a more fundamental goal and is more flexible for applications, but such experiments were not conducted in the manuscript. For example, if we want to put more information into the watermark than the current Tree-ring (e.g., a tree-ring watermark with high information density, like a circular barcode), then good noise reconstruction will be more important. To show this, many papers compare MSE or NMSE for noise reconstruction. Without this comparison, and based on my research experience and personal belief, it is hard to believe that FARI got good results in Table 1 because it is good at noise reconstruction. I suggest adding another table, like Table 1, that shows the round-trip MSE loss for noise reconstruction and image reconstruction. Also, it would be good to add qualitative results, not just quantitative ones. For example, pass the estimated noise $\hat{z_T}$ through the Decoder and compare it to the original $z_T$, and add error maps. Honestly, I think this new Table and Figure showing the inversion error directly is more important than the current Table 1.

2. The models used for experiments are not the latest. Rectified Flow is a more advanced diffusion model with reduced curvature, and I understand that SD3 (the successor to SD 1.5 and SD 2.1) is a Rectified Flow model. For practical use, it would be good to see experiments on models like SD3.

3. L231 states it is memory-efficient because of LoRA. But, it will use the same amount of RAM even if you train a separate network. It should be called disk-efficient because it saves some neural network weights. But saving a little disk space on SD weights is not a big advantage.

4. Overall, the font size in the figures is small. Also, some subsections are just one big paragraph, which makes it hard to read. Figure 2 could be made larger by increasing its width. For tables, there are better formats like https://github.com/jonbarron/tabilize/blob/main/tabilize.ipynb. This is used often and would be good to consider.

**Questions:**

If there are any changes from the original settings proposed in the Tree-rings watermark paper, please summarize them.

---

> ### Author Response · Authors · 2025-11-19
> **Rebuttal by Authors (1/2)**
>
> Thank you for your constructive feedback and detailed evaluation. We have addressed each of the points raised in our responses below.
>
> ---
>
> ## **Re: Concerns about performance (W1)**
>
> **Why is mathematical modeling not enough?**
>
> Yes, there are indeed many works that discuss inversion from a mathematical perspective, and we have compared with some of them in the Related Work section, appendix, and experimental section, such as BELM (which provides rigorous convergence guarantees). These methods demonstrate strong performance in image reconstruction or editing tasks. However, we argue that the problem setting in watermarking scenarios differs substantially, with two critical aspects being underestimated or overlooked by these mathematical approaches:
>
> 1. **Unknown generation conditions**: The conditions used for image generation, including guidance scale and prompt, are unavailable during inversion. This represents a fundamental information loss that these methods do not explicitly model.
> 2. **More importantly, distortion during transmission**: In the watermarking context, images undergo propagation through channels after generation and may suffer various distortions or attacks. These methods do not model the potential distortions that images may encounter.
>
> **Why do inversion for watermark extraction and image reconstruction differ?**
>
> For image reconstruction tasks, the goal is high-fidelity editing where the system takes an image as input and produces a similar image as output. In essence, these methods invert $z_0$ (potentially from $z_T$) to obtain $z'_T$, then reconstruct $z'_0$ from $z'_T$, optimizing for similarity between $z_0$ and $z'_0$. Obtaining an accurate $z'_T$ facilitates this objective, though it is not strictly necessary.
>
> In contrast, naive noise reconstruction performs the reverse process: transforming $z_T$ to $z_0$, then inverting $z_0$ back to $z'_T$. **However**, in practical watermarking scenarios, there are at least two systems with distinct responsibilities and information asymmetry.
>
> The generation process ($z_T \to z_0$) is performed by the user (Alice) with certain generation conditions known only to her. Subsequently, when watermark verification is needed, the inversion process ($z_0 \to z'_T$) is conducted by the verifier (Bob), who has no access to Alice's generation conditions.
>
> The situation becomes more complex in practice: the image Bob receives for verification may not originate entirely from Alice. Propagation through social media introduces noise to the watermarked images.
>
> In image reconstruction tasks, the environment can be idealized to facilitate systematic mathematical analysis. However, FARI operates under the realistic constraints of watermarking scenarios, where such idealization is not applicable.
>
> **Why learning-based method?**
>
> When external distortions must be considered, we find that the impact of internal distortions becomes relatively minor—as you can see, watermark extraction on clean images consistently performs well. We believe that modeling noise in complex scenarios using pure mathematical approaches is challenging. Therefore, employing learning-based methods for reconstruction is justified, as their ability to model real-world noise has been extensively validated. Furthermore, we call it FARI (Fast Asymmetric Robust Inversion), aiming for **robust inversion** rather than **exact inversion** like previous works.
>
> **Distortion settings**
>
> The distortion intensities used in our experiments are consistent with those in the Gaussian Shading paper. We believe they are non-trivial (especially for Tree-Ring) and sufficient to demonstrate the gap between FARI and other methods.
>
> **Evaluation metrics**
>
> Since FARI is specifically designed for watermark extraction, directly presenting downstream results helps readers better understand our contribution. Therefore, we primarily report FARI's performance on downstream watermarking methods in the paper. Your suggestion indeed helps us better showcase our advantages, so we have also included quantitative MSE metrics (Table 7) and qualitative visual results (Figure 10) in the revised paper.

---

> > ### Author Response · Authors · 2025-11-19
> > **A simple diagram illustrating the differences.**
> >
> > We hope the diagram below helps clarify the differences:
> > ```
> > +---------------------------------------+    +---------------------------------------+
> > |  1. Image Reconstruction              |    |  2. Naive Noise Reconstruction        |
> > +---------------------------------------+    +---------------------------------------+
> > |                                       |    |                                       |
> > |   z0 (Input Image)                    |    |   zT (Input Noise)                    |
> > |       |                               |    |       |                               |
> > |       v                               |    |       v                               |
> > |   [ Invert to z'T ]                   |    |   [ Transform to z0 ]                 |
> > |       |                               |    |       |                               |
> > |       v                               |    |       v                               |
> > |      z'T                              |    |      z0                               |
> > |       |                               |    |       |                               |
> > |       v                               |    |       v                               |
> > |   [ Reconstruct z'0 ]                 |    |   [ Invert to z'T ]                   |
> > |       |                               |    |       |                               |
> > |       v                               |    |       v                               |
> > |   z'0 (Output Image)                  |    |      z'T                              |
> > |       ^                               |    |                                       |
> > |       | (Optimize Similarity)         |    |                                       |
> > |       |                               |    |                                       |
> > +---------------------------------------+    +---------------------------------------+
> >
> > +-----------------------------------------------------------------------------------+
> > |  3. Watermarking Scenario (FARI - Realistic Constraints)                          |
> > +-----------------------------------------------------------------------------------+
> > |   [ ALICE ] (User/Generator)                                                      |
> > |   +-------------------+                                                           |
> > |   | zT (Latent Noise) |                                                           |
> > |   +---------+---------+                                                           |
> > |             |                                                                     |
> > |             v                                                                     |
> > |      [ Generation ] <--- (Private Conditions known ONLY to Alice)                 |
> > |             |                                                                     |
> > |             v                                                                     |
> > |     z0 (Watermarked Img)                                                          |
> > |             |                                                                     |
> > |   ..........|..................................................................   |
> > |   .         v                                                                 .   |
> > |   .    [ The Wild / Real World ]                                              .   |
> > |   .    - Social Media Propagation (Noise)                                     .   |
> > |   .         |                                                                 .   |
> > |   ..........|..................................................................   |
> > |             |                                                                     |
> > |             v                                                                     |
> > |   [ BOB ] (Verifier)                                                              |
> > |   +---------+---------+                                                           |
> > |   | Received Image    |                                                           |
> > |   +---------+---------+                                                           |
> > |             |                                                                     |
> > |             v                                                                     |
> > |      [ Inversion ]  <--- (Information Asymmetry: Bob has NO Access to Conditions) |
> > |             |                                                                     |
> > |             v                                                                     |
> > |            z'T                                                                    |
> > |             |                                                                     |
> > |             v                                                                     |
> > |      [ Verification ]                                                             |
> > +-----------------------------------------------------------------------------------+
> > ```

---

> > > ### Comment · Reviewer_8NsM · 2025-11-23
> > >
> > > Thanks for your concrete and well-written rebuttal.
> > >
> > > I am quite satisfied that the MSE difference (Tab 7 and Fig 10) is substantial and clear. This is concrete evidence that overturns my previous suspicion: that the model was fundamentally skipping Robust inversion and only performing well on the final task of watermark detection. The experimental results on SD3 (W2) also serve as good evidence, showing that the successful watermark detection was not achieved merely by chance. Therefore, I believe the issues I pointed out as W1/W2 have been addressed to a certain extent.
> > >
> > > My major remaining concern is: Bob must know the specific setting (e.g., which model did Alice use) and have well-trained inverters for every candidate models (e.g., SD1.5, SD2.1, SD3, and so on), while just applying Naive DDIM inversion doesn't.

---

> > > > ### Author Response · Authors · 2025-11-23
> > > >
> > > > Thank you for your response and for recognizing our work.
> > > >
> > > > Regarding the new concerns you raised, we decompose them into two parts: first, Bob must know which model Alice used in order to perform extraction; second, Bob must prepare an inverter for all models.
> > > >
> > > > **For the first concern**, we believe this limitation stems from inversion-based methods themselves, rather than being introduced by FARI. **Even with naive DDIM inversion, the corresponding model weights are required for successful and accurate extraction.** For instance, an image generated by Alice using SD2.1 cannot have its watermark extracted through inversion with SD3 weights—this issue exists regardless of whether FARI is used. FARI focuses on enhancing the efficiency and robustness of existing schemes. We also believe that constructing a unified inversion extractor for all models could be another very promising research direction.
> > > >
> > > > **For the second concern**, as mentioned above, even with simple DDIM inversion alone, Bob already needs to prepare model weights for various versions. The additional requirement of preparing FARI's LoRA weights does not impose a significant burden, while offering substantial benefits for three reasons:
> > > >
> > > > 1. Watermark extractors or extraction methods are typically provided by the watermark provider, who trains once and distributes to multiple users. Bob, as the verifier, does not bear the training cost.
> > > > 2. While training-free methods are indeed elegant, FARI's training cost is extremely low. Achieving simultaneous improvements in both efficiency and robustness with minimal training is worthwhile.
> > > > 3. You may notice that FARI demonstrates exceptional performance in certain scenarios. For example, Tree-Ring on SD3 is nearly unusable with naive inversion when suffering distortions, but achieves excellent performance with FARI. In other words, FARI can expand the applicability of such watermarking methods.

---

> ### Author Response · Authors · 2025-11-19
> **Rebuttal by Authors (2/2)**
>
> We present a subset of the MSE results here (on SD v1.5) for your reference. FARI demonstrates strong performance on the MSE metric as well.
>
> | Method | NFE | Clean | Adv. | JPEG | R.Crop | R.Drop | Resize | G.Blur | M.Blur | G.Noise | S&P | Bright |
> | --- | --- | --- | --- | --- | --- | --- | --- | --- | --- | --- | --- | --- |
> | DDIM | 50 | **0.2003** | 0.9285 | 0.8740 | 0.8734 | 0.8834 | 0.7671 | 0.9706 | 0.6920 | 1.1860 | 1.2963 | 0.8139 |
> | DDIM | 1 | 0.7108 | 0.9016 | 0.8981 | 0.9646 | 0.9621 | 0.8294 | 0.8987 | 0.8157 | 0.9067 | 0.9316 | 0.9074 |
> | EDICT | 50 | 0.2646 | 1.0477 | 1.0400 | 1.0407 | 1.0547 | 0.8569 | 1.0756 | 0.7654 | 1.2852 | 1.3735 | 0.9370 |
> | BELM | 50 | 0.6593 | 1.1723 | 1.0230 | 1.4621 | 1.4384 | 0.9181 | 1.1744 | 0.8789 | 1.2193 | 1.2065 | 1.2305 |
> | AMED† | 2 | 0.3577 | 0.8375 | 0.8022 | 0.8756 | 0.8800 | 0.7116 | 0.8733 | 0.6439 | 0.9562 | 1.0033 | 0.7917 |
> | LCM-LoRA | 2 | 0.4488 | 0.8471 | 0.7773 | 0.8264 | 0.8331 | 0.7296 | 0.8614 | 0.7126 | 1.0413 | 1.0904 | 0.7517 |
> | DMD2 | 1 | 0.7934 | 0.9201 | 0.9478 | 0.9472 | 0.9392 | 0.8673 | 0.8999 | 0.8578 | 0.9367 | 0.9587 | 0.9266 |
> | FARI | 1 | 0.2033 | **0.6699** | **0.6138** | **0.7385** | **0.7396** | **0.5650** | **0.7037** | **0.5486** | **0.7146** | **0.7920** | **0.6135** |
>
> ---
>
> ## **Re: Performance on SD 3 (W2)**
>
> Following your suggestion, we evaluate FARI's performance on SD v3.5 Medium (https://huggingface.co/stabilityai/stable-diffusion-3.5-medium). Since SD v3.5 Medium uses FlowMatchEulerDiscreteScheduler as the sampler and does not support DDIM, we implement inversion for this model following the approach in [1].
>
> To better demonstrate the improvements brought by FARI, we adopt more aggressive watermarking settings. Given that SD v3.5 has 16 latent channels, we set $f_{ch}=2$ and $f_h=f_w=8$ for Gaussian Shading, resulting in a watermark capacity of 512 bits. For Tree-Ring Watermark, we use only the last four channels for watermark embedding. Following your suggestion, we also report MSE results. The corresponding results are shown in the tables below, demonstrating FARI's strong generalization capability as it outperforms the 20-step naive inversion.
>
> GS Bit. Acc.:
>
> | Method | NFE | Clean | Adv. | JPEG | R.Crop | R.Drop | Resize | G.Blur | M.Blur | G.Noise | S&P | Bright |
> | --- | --- | --- | --- | --- | --- | --- | --- | --- | --- | --- | --- | --- |
> | Naive | 20 | 0.9994 | 0.9178 | 0.9440 | 0.9654 | 0.9403 | 0.9437 | 0.8728 | 0.9609 | 0.7778 | 0.8736 | 0.9818 |
> | FARI | 1 | 0.9995 | 0.9515 | 0.9671 | 0.9713 | 0.9469 | 0.9756 | 0.9384 | 0.9852 | 0.8765 | 0.9191 | 0.9833 |
>
> TR TPR:
>
> | Method | NFE | Clean | Adv. | JPEG | R.Crop | R.Drop | Resize | G.Blur | M.Blur | G.Noise | S&P | Bright |
> | --- | --- | --- | --- | --- | --- | --- | --- | --- | --- | --- | --- | --- |
> | Naive | 20 | 0.958 | 0.161 | 0.020 | 0.364 | 0.543 | 0.000 | 0.000 | 0.001 | 0.009 | 0.003 | 0.512 |
> | FARI | 1 | 0.998 | 0.978 | 0.994 | 0.998 | 0.995 | 0.995 | 0.952 | 0.996 | 0.917 | 0.968 | 0.985 |
>
> MSE:
>
> | Method | NFE | Clean | Adv. | JPEG | R.Crop | R.Drop | Resize | G.Blur | M.Blur | G.Noise | S&P | Bright |
> | --- | --- | --- | --- | --- | --- | --- | --- | --- | --- | --- | --- | --- |
> | Naive | 20 | 0.5632 | 1.1872 | 1.3983 | 0.8966 | 0.9079 | 1.2350 | 1.1638 | 1.2709 | 1.8126 | 1.7478 | 0.8889 |
> | FARI | 1 | 0.5108 | 0.8364 | 0.8624 | 0.8293 | 0.8447 | 0.8840 | 0.9386 | 0.8657 | 0.9302 | 0.9264 | 0.7360 |
>
> [1] T2SMark: Balancing Robustness and Diversity in Noise-as-Watermark for Diffusion Models, NeurIPS 2025
>
> ---
>
> ## **Re: Memory efficiency (W3)**
>
> Without using LoRA, supporting both image generation and watermark extraction simultaneously requires loading and maintaining two SD models with identical architectures but different parameters in memory, resulting in nearly double the GPU memory consumption. Hot-swapping LoRA modules alleviates this issue. Of course, the disk storage advantage you mentioned also exists.
>
> For SD v2.1 at fp16 precision, loading two models (sharing VAE and text encoder) requires approximately 7889 MB of GPU memory, while using LoRA only requires 4027 MB. As model size increases, this advantage becomes even more pronounced.
>
> ---
>
> ## **Re: Formatting issues (W4)**
> Thank you for your suggestions. We have adjusted the size of figures, including Figure 2, and used highlighting to emphasize key content. We hope these improvements enhance readability.
>
> ---
>
> ## **Re: Tree-Ring settings (Q1)**
> We provide detailed experimental settings in the Appendix B. Specifically, we use Tree-Ring in its `rand` mode, with the watermark embedded in the fourth channel of the latent space.

---

### Official Review · Reviewer_vuw4 · 2025-10-30

**Soundness:** 3
**Presentation:** 3
**Contribution:** 3
**Rating:** 6
**Confidence:** 3

**Summary:**

This work builds off the literature on inversion-based watermarking that use diffusion forward and backward processes to encode and then recover watermarks. The authors find a critical weakness in many watermarking approaches: the inversion process which is both weak and slow. They then propose a methodology that uses LoRA and an adversarial suite to not only make their watermarking more robust but much faster.

**Strengths:**

1. The approach identifies a key weakness in existing watermarking literature.
2. I believe that the approach is new to my knowledge, particularly the insights about curvature and low-NFE approximation.
3. One step inversion is commonly done (DDIM) but the authors note that these approaches have limitations.
4. Empirical results seem promising

**Weaknesses:**

1. Existing works commonly use models such as SDXL-Turbo (one-step generation) to achieve faster watermarking speeds [1]. I believe that these methods may be faster and have potentially better quality.
2. We should compare against some more SOTA attacks (regeneration, combination of many attacks)
3. The concept of using an adversarial testing suite + introducing trainable parameters is not that new.
4. Requires training (weakness compared to DDIM which seems very good in its base version looking at Table 1). Given the simplicity and existing speed benefit (of DDIM single step) I find it a bit harder to motivate the use of this method.


[1] Lu, Shilin, et al. "Robust watermarking using generative priors against image editing: From benchmarking to advances." arXiv preprint arXiv:2410.18775 (2024).

**Questions:**

1. What is the performance relative to VINE?
2. Please try attaching Gaussian Shading + Tree-Ring with SDXL-Turbo backbone. I wonder about the robustness/speed/image quality comparison.
3. I wonder about the generalizability as we seem to test/train on a singular dataset. I think that there is a need to explore other kinds/styles of images.

---

> ### Author Response · Authors · 2025-11-19
> **Rebuttal by Authors (1/3)**
>
> Thank you for your time and valuable feedback on our submission. We have carefully considered each point raised and address them individually below.
>
> ---
>
> ## **Re: Comparison with VINE (W1 & Q1)**
>
> VINE is an excellent work that leverages SD-Turbo as a watermark encoder and demonstrates strong resilience against editing operations. However, it differs fundamentally from the inversion-based watermarking paradigm that our paper focuses on. VINE is a post-processing watermark, meaning it adds watermarks to already-generated images, whereas Gaussian Shading (GS) and Tree-Ring (TR) are inversion-based generative watermarks that embed watermarks during the image generation process. Both approaches show great promise: post-processing watermarks offer stronger generalizability but require training, while GS and TR are training-free methods.
> To compare their performance, we evaluate VINE against FARI-enhanced Gaussian Shading (GS) and Tree-Ring (TR) using 1,000 samples from the MS-COCO dataset across three dimensions: efficiency, image quality, and robustness. For efficiency, we directly compare the embedding and extraction time of all watermarking methods. For image quality, since generative watermarks cannot be evaluated using relative metrics like PSNR, we employ generative image quality metrics for fair comparison. For robustness, we use the same evaluation metrics as presented in the main paper.
> For VINE, we use the official repository code and pre-trained weights. Since VINE-Robust demonstrates stronger robustness compared to VINE-Base, we select it for comparison.
> The corresponding results are as follows:
>
> Bitwise Accuracy:
>
> | Method(Capacity) | Clean | Adv. | JPEG | R.Crop | R.Drop | Resize | G.Blur | M.Blur | G.Noise | S&P | Bright |
> | --- | --- | --- | --- | --- | --- | --- | --- | --- | --- | --- | --- |
> | VINE (100 bits) | 0.9997 | 0.8884 | 0.9981 | 0.5134 | 0.9970 | 0.9596 | 0.9667 | 0.9985 | 0.8666 | 0.9870 | 0.7087 |
> | GS-FARI (256 bits) | 1.0000 | 0.9835 | 0.9944 | 0.9781 | 0.9774 | 0.9994 | 0.9954 | 0.9997 | 0.9762 | 0.9655 | 0.9656 |
>
> TPR@1e-6:
>
> | Method | Clean | Adv. | JPEG | R.Crop | R.Drop | Resize | G.Blur | M.Blur | G.Noise | S&P | Bright |
> | --- | --- | --- | --- | --- | --- | --- | --- | --- | --- | --- | --- |
> | VINE | 1.000 | 0.802 | 1.000 | 0.006 | 1.000 | 0.998 | 1.000 | 1.000 | 0.842 | 1.000 | 0.372 |
> | GS-FARI  | 1.000 | 0.999 | 0.999 | 1.000 | 1.000 | 1.000 | 0.999 | 0.999 | 1.000 | 1.000 | 0.991 |
> | TR-FARI | 1.000 | 0.992 | 0.998 | 1.000 | 1.000 | 1.000 | 1.000 | 1.000 | 0.949 | 0.999 | 0.981 |
>
> Efficiency and image quality:
>
> | Method | Embedding Time/s | Extraction Time/s | CLIP Score↑ | FID↓ |
> | --- | --- | --- | --- | --- |
> | VINE | 0.0738 | 0.0103 | 0.3437 | 52.80 |
> | GS-FARI | 0.0005 | 0.0804 | 0.3450 | 53.44 |
> | TR-FARI | 0.0008 | 0.1062 | 0.3453 | 54.29 |
>
> The results show that each method has its own strengths and weaknesses. We observe that VINE may be vulnerable to specific distortions, such as random cropping and brightness adjustments, which could be attributed to its training strategy.

---

> ### Author Response · Authors · 2025-11-19
> **Rebuttal by Authors (2/3)**
>
> ## **Re: Robustness against various attacks (W2)**
>
> We evaluate three different regeneration attacks. For [1], we set the noise steps to 300. For [2] and [3], we set the quality parameter to 3 (representing the highest attack strength). The corresponding results are as follows:
>
> GS Bit. Acc.:
>
> | Attack | DDIM (50 steps) | FARI (1 step) |
> | --- | --- | --- |
> | diff_attacker_300 [1] | 0.9500 | 0.9533 |
> | cheng2020-anchor_3 [2] | 0.9859 | 0.9896 |
> | bmshj2018-factorized_3 [3] | 0.9875 | 0.9904 |
>
> TR TPR:
>
> | Attack | DDIM (50 steps) | FARI (1 step) |
> | --- | --- | --- |
> | diff_attacker_300 [1] | 0.516 | 0.660 |
> | cheng2020-anchor_3 [2] | 0.998 | 0.999 |
> | bmshj2018-factorized_3 [3] | 0.982 | 0.997 |
>
> The results show that FARI provides improvements in robustness. However, the ability to resist attacks remains largely dependent on the underlying design of the base watermarking methods, as FARI serves as a plug-and-play inversion approach to enhance their detection performance.
>
> [1] Invisible Image Watermarks Are Provably Removable Using Generative AI, NeurIPS 2024
>
> [2] Learned Image Compression with Discretized Gaussian Mixture Likelihoods and Attention Modules, CVPR 2020
>
> [3] Variational image compression with a scale hyperprior, ICLR 2018
>
> ---
>
> ## **Re: About Novelty (W3)**
>
> While adversarial training is indeed common in watermarking-related work, applying it to enhance inversion is non-trivial.
>
> The primary goal of this paper is to simultaneously improve the extraction efficiency and robustness of inversion-based watermarks through a novel inversion method—a pressing problem that urgently needs to be addressed. To this end, we analyze the limitations of existing inversion methods and identify that they fail to account for external distortions, causing their acceleration or accuracy improvement mechanisms to become ineffective. As mentioned in the introduction, adversarial training is a natural approach. However, the challenge lies in the fact that multi-step inversion incurs prohibitively high backpropagation costs, while few-step inversion may compromise accuracy. This naturally leads us to distillation techniques. While this direction is conceptually straightforward, the key insight is that current widely-used high-precision trajectory distillation techniques for diffusion models require extremely high training costs, making distillation an impractical solution for accelerating watermark extraction. This is where our discovery regarding curvature plays a crucial role: it provides the foundation for fast and effective distillation, enabling us to propose the FARI method.
>
> ---
>
> ## **Re: Limited improvement (W4)**
>
> While single-step DDIM offers advantages in speed, its performance degradation cannot be ignored, preventing practical deployment. Here is the evidence:
>
> - For Gaussian Shading: The improvement FARI brings over naive single-step DDIM would require 2-3× redundancy if achieved through alternative means, representing a substantial sacrifice in watermark capacity. This improvement is far from trivial.
> - For Tree-Ring Watermark: Single-step DDIM achieves only <90% TPR, which is insufficient for practical use, whereas FARI approaches nearly 100%.
> - For MSE: We have also added Table 7 measuring MSE and visual results of reconstructed noise (Figure 10) to the paper. These additions will help better illustrate FARI's performance advantages.
>
> FARI is designed to enable fast inversion without sacrificing robustness. In fact, as demonstrated in our results, FARI even enhances robustness. Moreover, FARI's training is computationally efficient while delivering considerable performance gains.

---

> ### Author Response · Authors · 2025-11-19
> **Rebuttal by Authors (3/3)**
>
> ## **Re: Performance on SDXL-Turbo backbone (Q2)**
>
> We conduct experiments on SDXL-Turbo using DDIMScheduler for one-step generation and inversion. Since SDXL-Turbo does not support classifier-free guidance (CFG), the guidance scale is set to 0. All other training and testing settings remain consistent with those in the main paper. The comparative results are shown in the table below.
>
> GS Bit. Acc.:
>
> | Method | Clean | Adv. | JPEG | R.Crop | R.Drop | Resize | G.Blur | M.Blur | G.Noise | S&P | Bright |
> | --- | --- | --- | --- | --- | --- | --- | --- | --- | --- | --- | --- |
> | DDIM | 0.9755 | 0.8576 | 0.9142 | 0.7449 | 0.7552 | 0.9370 | 0.8841 | 0.9346 | 0.8935 | 0.8229 | 0.8315 |
> | FARI | 0.9998 | 0.9511 | 0.9903 | 0.9115 | 0.9161 | 0.9871 | 0.9370 | 0.9922 | 0.9801 | 0.9354 | 0.9102 |
>
> TR TPR:
>
> | Method | Clean | Adv. | JPEG | R.Crop | R.Drop | Resize | G.Blur | M.Blur | G.Noise | S&P | Bright |
> | --- | --- | --- | --- | --- | --- | --- | --- | --- | --- | --- | --- |
> | DDIM | 0.561 | 0.201 | 0.672 | 0.000 | 0.000 | 0.065 | 0.001 | 0.221 | 0.412 | 0.271 | 0.163 |
> | FARI | 0.998 | 0.855 | 0.967 | 0.743 | 0.933 | 0.893 | 0.683 | 0.928 | 0.949 | 0.837 | 0.763 |
>
> FARI demonstrates substantial improvement over DDIM. In terms of speed, since SDXL-Turbo itself performs single-step inversion, both methods achieve extraction times around 0.2235s. While there are indeed many single-step generation models currently available, our goal is to enable multi-step generation models to achieve comparable efficiency with single-step watermark extraction.
>
> The generation quality is primarily influenced by the watermarking method itself. We present our measurements here using unwatermarked generation results as the baseline.
>
> | Method | CLIP Score↑ | FID↓ |
> | --- | --- | --- |
> | SDXL-Turbo | 0.3537 | 59.78 |
> | GS-FARI | 0.3547 | 59.25 |
> | TR-FARI | 0.3539 | 59.65 |
>
> ---
>
> ## **Re: Results on more datasets (Q3)**
>
> Thank you for the suggestion. In the original paper, we trained on the COCO dataset and test on the SDP dataset. To further validate generalization across datasets, we additionally select 1,000 prompts from a new dataset (DiffusionDB, https://huggingface.co/datasets/poloclub/diffusiondb). Combined with the existing SDP and COCO datasets, we conduct cross-dataset validation. The results are shown below, presented in the format of TR TPR / GS Bit Acc, with 50-step DDIM as the baseline.
>
> | Training dataset \ Test dataset | SDP | COCO | DiffusionDB |
> | --- | --- | --- | --- |
> | DDIM Baseline (50-step) | 0.957 / 0.9780 | 0.948 / 0.9785 | 0.938 / 0.9763 |
> | SDP | \ | 0.993 / 0.9852 | 0.990 / 0.9839 |
> | COCO | 0.994 / 0.9841 | \ | 0.992 / 0.9846 |
> | DiffusionDB | 0.994 / 0.9832 | 0.994 / 0.9861 | \ |

---

> > ### Comment · Reviewer_vuw4 · 2025-11-27
> > **Response**
> >
> > I thank the authors for their response. The effort put into this rebuttal is very comprehensive and detailed. I will retain my positive review of this paper. I think that this contribution is quite nice :)

---

> > > ### Author Response · Authors · 2025-11-28
> > >
> > > Thank you for recognizing our work and rebuttal, and for your positive evaluation!

---

### Comment · Area_Chair_n3XJ · 2025-11-25
**Authors' responses**

Dear Reviewers,

The authors have submitted their responses to your questions and feedbacks. Please read them and give your comments.

Regards,
AC

---

### Comment · Area_Chair_n3XJ · 2025-11-28
**The Author/Reviewer Discussion Phase deadline is approaching**

Dear Reviewers,

The Author/Reviewer Discussion Phase deadline is approaching. If you have not responded to authors’ rebuttal, please read it and give your feedback asap.

Regards,
AC

---

### Author Response · Authors · 2025-12-01
**Summary of Discussion**

We thank the Area Chairs and Reviewers for their efforts in building a fair academic community and helping us improve our work!

### **First, we aim to clarify our contributions more explicitly:**

Inversion speed and robustness represent two critical challenges in the practical deployment of inversion-based watermarking, frequently cited as limitations and future work in existing literature. FARI provides a simple yet effective approach that simultaneously addresses both challenges, breaking the commonly perceived trade-off between speed and robustness. Specifically, in this paper, we make the following contributions:

1. We identify the convergence failure problem of pure mathematical inversion in watermarking scenarios due to external distortions.
2. We present a key insight: the inversion trajectory of diffusion models exhibits lower curvature than denoising, which facilitates distillation to compress NFE and enables efficient adversarial training.
3. We propose FARI, a simple yet effective method that achieves robustness surpassing 50-step Naive DDIM inversion with just one-step inversion, requiring only 20 minutes of fine-tuning on a single NVIDIA A6000 GPU (for Stable Diffusion v2.1). This represents a speedup of several dozen times.

### **We then summarize our efforts in this rebuttal and the corresponding feedback from the reviewers:**

In the original submission, we have included extensive generalization and ablation experiments in both the main text and appendix. Following the Reviewers' suggestions, we further validated FARI's performance across:

- **Different models**: SD v3.5 (Table 5) and SDXL-Turbo (Table 6) — requested and acknowledged by Reviewers vuw4 and 8NsM
- **Different datasets**: Cross-validation across three datasets (Table 7) — requested and acknowledged by Reviewer vuw4
- **Different attacks**: Three regeneration attacks and one optimization attack (Figure 10) — requested and acknowledged by Reviewers vuw4 and yTFb
- **Different hyperparameters**: Different choices of $t$ in Eq. 9 (Table 10) — requested and acknowledged by Reviewer yTFb
- **Different metrics**: TPR@1e-3 for all experiments — requested and acknowledged by Reviewer yTFb; Mean Square Error (Table 11 and Figure 11) — requested and acknowledged by Reviewer 8NsM

**All reviewers expressed satisfaction with the results of the additional experiments they requested.** We have also addressed the detailed clarifications requested by Reviewer yTFb, including the role of LoRA, the choice of timestep t, and details regarding the curvature measurement experiments. All revisions have been incorporated into the latest manuscript and are highlighted in blue. We believe these revisions have adequately addressed all concerns and that our paper now meets the acceptance criteria.

During the discussion phase, **we reached consensus with three reviewers** (vuw4, 3yip and yTFb) who all acknowledged our rebuttal:

- Reviewers vuw4 and 3yip maintained their positive scores (6)
- Reviewer yTFb expressed satisfaction with our responses and revisions, raising their score from 4 to 8.

Although we did not receive a final response from Reviewer 8NsM, in the completed discussions, **they expressed strong satisfaction with our first-round response and new results (addressing their main concerns from the review phase that led them to consider rejection)**. We also provided detailed answers to their follow-up questions and believe these sufficiently address their concerns.

Overall, **our contributions and rebuttal have been widely recognized by the reviewers**, and we hope the AC will take this into consideration when making the final decision.

Thanks again for the Area Chairs' and Reviewers' efforts!

---

### Meta-Review · Area_Chair_PfF7 · 2026-01-07

**Summary:**

Reviewer 8NsM questioned whether learning-based inversion can achieve good reconstruction fidelity given the information asymmetry between forward and backward processes, requesting MSE metrics rather than just downstream watermark performance. Reviewer yTFb flagged methodological issues: extrapolating TPR to 1e-6 from only 1000 samples is questionable, the timestep t=0 choice was not justified, and LoRA's role was unclear. Multiple reviewers noted the technical components (distillation, adversarial training, LoRA) are established, with novelty residing in the curvature observation and their integration. Reviewer 3yip raised concerns about mismatch between unconditional inversion and CFG-guided generation.

**Reviewer Concerns:**

The authors addressed several concerns: MSE experiments were added showing FARI outperforms baselines, which Reviewer 8NsM acknowledged overturned their suspicion. TPR was recalculated at 1e-3 FPR which is more defensible and led to yTFb raising their score; an ablation on timestep t justified the t=0 choice; LoRA's dual role (enabling distillation and robustness) was clarified; and experiments on SD3.5 and SDXL-Turbo demonstrated generalization. Some issues remain partially outstanding. Reviewer 8NsM raised a practical concern that verifiers need model-specific trained inverters, adding deployment burden. The authors argue this applies to all inversion-based methods, which is fair, but FARI does add a training artifact per model. The CFG mismatch is mitigated empirically but not resolved in a principled way. I consider them as limitations rather than critical flaws that would prevent the paper from getting accepted. I think the author's rebuttal was strong and convincing.

**Reviewer Scores:**

Reviewer vuw4 (6) was satisfied and would stay at 6. Reviewer 8NsM (2) acknowledged the MSE results addressed their weaknesses "to a certain extent", but raised a new concern about the need for model-specific inverters which the authors address in their follow-up rebuttal. I am convincend by the author's argumentation. Reviewer 3yip (6) maintained their score after brief exchange and reviewer yTFb increased their score from 4 to 8 after their concerns about evaluation protocol and method clarity were addressed. Overall, I recommend acceptance of this paper.

---

### Decision · Program_Chairs · 2026-01-26

Accept (Poster)